



**Distributed *vs*. semi-distributed simulations of snowpack**
**dynamics in alpine areas: case study in the upper Arve**
**catchment, French Alps, 1989–2015**
Jesús Revuelto[1,2], Grégoire Lecourt[1], Matthieu. Lafaysse[1], Isabella Zin[2], Luc Charrois[1], Vincent Vionnet[1],
Marie Dumont[1], Antoine Rabatel[2], Delphine Six[2], Thomas Condom[2], Samuel Morin[1], Alessandra Viani[2,3],
Pascal. Sirguey[4]
[1] Météo-France - CNRS, CNRM, UMR 3589, CEN, Grenoble, France
[2] Université Grenoble Alpes, CNRS, IRD, Institut des Géosciences de l'Environnement (IGE, UMR
5001), Grenoble, France
[3] University of Brescia, Department of Civil Engineering, Architecture, Land, Environment and
Mathematics (DICATAM), Brescia, Italy
[4] National School of Surveying, University of Otago, Dunedin, New Zealand
**Abstract:** We evaluated distributed and semi-distributed modeling approaches to
simulating the spatial and temporal evolution of snow and ice over an extended
mountain catchment, using the Crocus snowpack model. The distributed approach
simulated the snowpack dynamics on a 250-m grid, enabling inclusion of terrain
shadowing effects. The semi-distributed approach simulated the snowpack dynamics for
discrete topographic classes characterized by elevation range, aspect, and slope. This
provided a categorical simulation that was subsequently spatially re-projected over the
250-m grid used for the distributed simulations. The study area (the upper Arve
catchment, western Alps, France) is characterized by complex topography, including
steep slopes, an extensive glaciated area, and snow cover throughout the year.
Simulations were carried out for the period 1989–2015 using the SAFRAN
meteorological forcing system. The simulations were compared using four observation
datasets including point snow depth measurements, seasonal and annual glacier surface
mass balance, snow covered area evolution based on optical satellite sensors, and the
annual equilibrium-line altitude of glacier zones, derived from satellite images. The
results showed that in both approaches the Crocus snowpack model effectively
reproduced the snowpack distribution over the study period. Slightly better results were
obtained using the distributed approach because it included the effects of shadows and
terrain characteristics.
**Key words:** snowpack simulation, distributed, semi-distributed, mountain areas,
glacierized catchments





## 1. Introduction

The dynamics of the accumulation and melting of snow and ice in mountain areas has
major effects on the timing and level of discharge from rivers in downstream areas.
One-sixth of the Earth's population depends directly on the water supply from snow and
ice melt in mountain areas (Barnett *et al.*, 2005). Thus, significant research effort has
been applied to the study of snow and ice dynamics in these regions (Egli and Jonas,
2009; Lehning *et al.*, 2011; López-Moreno *et al.*, 2013; McCreight *et al.*, 2012), with
particular focus on mountain hydrology (DeBeer and Pomeroy, 2009; López-Moreno
and García-Ruiz, 2004; Oreiller *et al.*, 2014; Viviroli *et al.*, 2007). The snowpack
dynamics and its spatial extent also control many mountain processes, including soil
erosion (Meusburger *et al.*, 2014), plant survival (Wipf *et al.*, 2009), and the glacier
surface mass balance (López-Moreno *et al.*, 2016; Réveillet *et al.*, 2017; Sold *et al.*,
46 2013).
Some of the most dangerous natural hazards in mountain areas are also directly related
to the distribution of the snowpack and ice, and their evolution over time. This is the
case for snow avalanches (Schweizer *et al.*, 2008), and floods in mountain rivers and
downstream areas (Gaál *et al.*, 2015). To enable anticipation of the occurrence of snow-
related hazards and to reduce the threat to populations and infrastructure (Berghuijs *et*
*al.*, 2016; Tacnet *et al.*, 2014); various models have been developed to reproduce and
forecast the evolution of the snowpack on a daily or sub-daily basis.
Detailed snowpack models (Bartelt and Lehning, 2002; Vionnet *et al.*, 2012) are
increasingly coupled with hydrological models to forecast river discharges, and this
depends on reliable simulation of snow and ice melting (Avanzi *et al.*, 2016; Braun *et*
*al.,* 1994; Lehning *et al.*, 2006). The more accurate the information on snowpack
dynamics, the better will be the discharge forecasts based on hydrological models.
However, the spatio-temporal distribution of the snowpack is highly variable in
mountain areas (López-Moreno *et al.*, 2011, 2013; Scipión *et al.*, 2013; Seidel *et al.*,
2016), and the runoff from mountain catchments depends on many interrelated
processes that are highly variable in space and time, including infiltration, surface
runoff, groundwater recharge, freezing of soil, and the snowpack distribution (Seyfried
and Wilcox, 1995). For example, in areas where snow persists throughout the year the
snowpack dynamics has a major impact on groundwater storage (Hood and Hayashi,



2015). Finally, snowpack models are also combined with other models and techniques
to forecast avalanche hazards (Bartelt and Lehning, 2002; Durand *et al.*, 1999).
Reproducing snowpack dynamics in heterogeneous mountain areas remains
challenging. Some snowpack processes, including wind-induced redistribution and
small scale topographic control on the snow distribution (Mott et al., 2010; Revuelto *et*
*al.*, 2016a; Schirmer *et al.*, 2011; Trujillo *et al.*, 2007; Vionnet et al., 2013) have not yet
been fully integrated into numerical snowpack models which can be used operationally.
Moreover, the additive nature of snowpack dynamics involves discrepancies between
observed and simulated snowpacks, which can accumulate over the simulation period
(e.g., Raleigh *et al.*, 2015).
The various approaches available for running snowpack simulations range from
punctual simulations (snowpack dynamics simulated for a particular location having
specific characteristics) to semi-distributed and distributed approaches that simulate
snow dynamics over broad areas.
The semi-distributed approach involves simulating the snowpack evolution over areas
defined using discrete values for topographic variables including altitude, aspect, and
slope (Fiddes and Gruber, 2012, 2014). The French numerical chain S2M (SAFRAN-
SURFEX-MEPRA; Lafaysse *et al.*, 2013), simulates the snowpack evolution using a
semi-distributed approach. In this chain the SURFEX/ISBA-Crocus snowpack model
(Vionnet *et al.*, 2012; hereafter referred to as Crocus) is applied over a semi-distributed
discretization of the French mountain ranges to diagnose the avalanche hazard for
various topographic classes. Semi-distributed hydrological simulations are also widely
used, which involves discretizing catchments into hydrologic response units (HRU),
with the flow contribution from the HRUs being routed and compounded into an overall
catchment discharge (Nester *et al.*, 2012; Pomeroy *et al.*, 2012). This simulation method
is also applied to river discharge forecasting in mountain areas, with the output of semi-
distributed snowpack simulations used as inputs to the hydrological models (Braun *et*
*al.,* 1994).
The other modeling approach to simulating snowpack dynamics over extended areas is
distributed simulations. This method involves simulation of the temporal evolution of
environmental variables (e.g., snowpack or other hydrological variables) over a gridded
representation of the terrain. In this approach the terrain is not discretized in classes;
rather, it explicitly considers the characteristics (e.g. elevation, slope, aspect) for each



pixel when simulating its snowpack evolution. Both approaches (distributed and semi-
distributed) have advantages and disadvantages, particularly the lower computing
resource requirements of semi-distributed simulations, and the more accurate terrain
representation of distributed simulations. Some snowpack processes cannot be
reproduced using the semi-distributed approach, including wind-induced snow
redistribution, small scale topographic control of precipitation, and terrain shadowing
effects (Grünewald *et al.*, 2010; Revuelto *et al.*, 2014; Vionnet *et al.*, 2014). However,
evaluating the performance of these simulation approaches depends on the intended use
of the simulations (Carpenter and Georgakakos, 2006; Orth *et al.*, 2015). Similarly, the
results obtained will depend on the spatial scale and the quality of the meteorological
forcing model, and whether it is distributed or semi-distributed (Queno *et al.*; 2016;
Vionnet *et al.*, 2016). Many studies have compared the performance of hydrological
models based on distributed and semi-distributed approaches in reproducing streamflow
dynamics for alpine watersheds (Grusson *et al.*, 2015; Kling and Nachtnebel, 2009; Li
*et al.*, 2015), but none have directly analyzed and compared representation of the spatio-
temporal evolution of the snowpack using these simulation approaches. This is
significant because direct implementation of the most promising advances in simulation
requires the use of distributed simulations. This is the case for assimilation of satellite
data (Charrois *et al.*, 2016; Dumont *et al.*, 2012a; Thirel *et al.*, 2013); the inclusion of
small scale processes in simulations, including snow redistribution by wind (Schirmer *et
al.*, 2011; Vionnet *et al.*, 2014); and gravitational or topographic controls on snow
movements (Bernhardt and Schulz, 2010; Christen *et al.*, 2010; Revuelto *et al.*, 2016a).
Thus, comparison of distributed and semi-distributed simulations is needed to evaluate
potential improvements, based on similar simulation setups (including the same study
period and area, meteorological forcing, and simulation initialization). The newest
meteorological models provide high spatial resolution information on the evolution of
atmospheric variables (Seity *et al.*, 2010); this is an improvement that distributed
snowpack simulations can fully incorporate.
This study provided a comprehensive evaluation of semi-distributed and distributed
snowpack simulations for a mountain catchment, using the Crocus snowpack model
(Brun *et al.*, 1992; Vionnet *et al.*, 2012). We firstly assessed the ability of the model to
simulate the snowpack evolution at a local scale for specific stations having continuous
snow observation data. For these stations, the punctual simulations accounted for local
topographic characteristics. These punctual simulations enabled initial analysis of the
capacity of the model to subsequently evaluate the distributed and semi-distributed
approaches to simulating the snowpack dynamics over a broader area, using the same
meteorological forcing. The simulation results obtained using the distributed and semi-
distributed approaches were compared with observations for the snow covered area
based on MODIS satellite sensors, the glacier surface mass balance (winter, summer,
and annual), and the glacier equilibrium-line altitude derived from satellite images
(Landsat, SPOT, and ASTER). This enabled assessment of the use of distributed
simulations for analysis of snow and ice dynamics. The simulations were based on data
for the upper Arve catchment (French Alps) for the 26 years from 1989 to 2015.



## 2. Study area

The upper Arve catchment is located in the western Alps, France, between the northeast slopes of the Mont Blanc massif and the southwest slopes of the Aiguilles Rouges massif. The catchment extends from the headwaters of the Arve River to the town of Chamonix (Fig. 1), and includes major tributaries carrying melt water from three glaciated areas (*Arveyron de la Mer de Glace, Arveyron d'Argentière, and Bisme du Tour*) to the main river. The upper Arve catchment covers 205 km$^2$ and has a high degree of topographic heterogeneity, with steep slopes in some areas, and gentle slopes on large glaciated areas and at the lower elevation zones of the valley, which is a typical U-shaped glacial valley. Elevation ranges from 1020 to 4225 m.a.s.l., with 65% of the surface area above 2000 m.a.s.l. Glaciers cover 33% of the area (Gardent *et al.*, 2014), and 22% is covered by forests, mainly in the lower elevation areas. The water discharge regime is strongly dependent on the snow melt dynamics during spring and early summer, with the major contribution of melt water from glacierized areas occurring during late summer and autumn; this is termed a nivo-glacial regime of river discharge (Viani *et al.*, submitted). The Mont Blanc and Aiguilles Rouges massifs are also highly spatially heterogeneous, having various slopes and aspects over a wide range of elevations in glaciated and non-glaciated areas; this affects the spatio-temporal evolution of snow and ice.

The area is one subject to severe flood hazards. This is a consequence of the steepness of the terrain, which results in a rapid hydrological response to precipitation, the typically rapid meteorological changes that occur in this mountain area (mainly associated with convective episodes during spring and summer), and the high population densities and infrastructure in the bottom of the valley.





## 3. Methods

### 3.1. Simulation setup

We used the Crocus snowpack model to simulate the temporal evolution of snow and ice in the upper Arve catchment. Crocus is a multilayer model that simulates snowpack evolution based on the energy and mass exchanges between the various snow layers within the snowpack, and between the snowpack and its interface with the atmosphere and the soil (i.e. the top and bottom of the snow column). The maximum number of layers in Crocus is set to 50. Crocus is implemented in the externalized surface model SURFEX (Vionnet *et al.*, 2012). Within SURFEX (Masson *et al.*, 2013), Crocus is coupled to the multilayer land surface model ISBA-DIF (Interaction between Soil, Biosphere and Atmosphere; diffusion version; Decharme *et al.*, 2011).

The meteorological forcing required to drive the temporal evolution of the simulations was obtained from the SAFRAN meteorological analysis system (Durand *et al.*, 1993). This provides the atmospheric variables needed to run ISBA-Crocus, including air temperature, specific humidity, long wave radiation, direct and diffuse short wave radiation, wind speed, and precipitation phase and rate. SAFRAN was specifically developed to provide meteorological forcing for mountain areas at a suitable elevational resolution. The SAFRAN analysis combines observational data obtained from automatic weather stations with manual observations with the guess from the global numerical weather prediction system ARPEGE (Courtier and Thépaut, 1994). We used SAFRAN re-analysis, which benefitted from meteorological observations not available in real time (Durand *et al.,* 2009a, 2009b). This analysis system can provide outputs for punctual simulations, or semi-distributed outputs. In the first case the analysis is performed directly for the elevations of the stations involved, while in the second case the analysis is performed for 300-m elevation bands. In both cases the spatial extent of the analysis is approximately 1000 km². These regions (known as "massifs") were defined by Durand *et al*. (1993) who took climatic homogeneity into account. In this study the SAFRAN analysis was only used for that part of the Mont Blanc "massif" which covers the entire study catchment. SAFRAN and SURFEX/ISBA-Crocus (hereafter SAFRAN-Crocus) are used in avalanche hazard forecasting in France, using the S2M chain (Lafaysse *et al.*, 2013); this takes account of the altitude, aspect, and slope classes (semi-distributed simulation).



3.2. Punctual, semi-distributed, and distributed approaches
The temporal evolution of snow and ice was simulated using punctual, semi-distributed,
and distributed approaches, based on the same meteorological forcing.
*Punctual simulation*
Punctual snowpack simulations were performed for the five Météo-France stations
within the study area, based on the elevation, slope, and aspect for each station.
Punctual simulations included a topographic mask from a 50-m digital elevation model
(DEM) to account for any terrain shadowing effect on simulation of the incoming
shortwave radiation (provided by the SAFRAN meteorological model).
*Semi-distributed simulation*
Snow and ice semi-distributed simulations were carried out based on the topographic
classes of the SAFRAN model (300-m elevation bands from 900 m.a.s.l. to 4100
m.a.s.l) for eight aspect classes (north, northeast, east, southeast, south, southwest, west,
and northwest) and two slope values (20° and 40°). For each elevation band a
simulation over flat terrain (no aspect classification) was also carried out. These
topographic classes are the same as those used for avalanche forecasting (Lafaysse *et*
*al.*, 2013). To consider snow and ice evolution on glacierized and non-glacierized areas,
two distinct simulations were run for all terrain classes, one involving a given thickness
of ice to initialize the simulation, and another initialized using bare ground (see section

218     3.3).

In a final stage the snowpack semi-distributed simulations were assigned or re-projected
onto the pixels of the study area DEM (the same DEM used for distributed simulations;
250x250 m grid size). The pixels were categorized according to the semi-distributed
terrain classes: slopes from 0 to 10° were considered flat, those from 11 to 30° were
assigned to the 20° slope class, and those > 30.1° were assigned to the 40° class. From
this categorization of the DEM the snowpack simulation outputs were assigned to each
terrain class for all time steps. Thereby, for each time step a snow and ice distribution
map was generated that spatially distributed the semi-distributed snowpack simulation
obtained for the various terrain classes. This enabled comparison of the two approaches
based on the same observation dataset.
*Distributed simulation*
The distributed snowpack simulations were performed in a DEM having a 250x250 m
grid spacing and covering the 205 km$^2$ of the study area. As SAFRAN reanalysis



provides semi-distributed outputs, the meteorological forcing at hourly time steps was
spatially distributed over the 250-m grid DEM using specific routines that accounted for
the topographic characteristics of each grid cell, based on interpolated meteorological
variables for the closest terrain classes (Vionnet *et al.*, 2016). However, the
meteorological model used was the same for all simulations, and only minor differences
occurred because of the need to include the topographic characteristics of each pixel.
The distributed Crocus simulations included the elevation, aspect, slope, soil, and land
cover characteristics for each pixel (the last two obtained from ECOCLIMAP-
II/Europe; Faroux *et al.*, 2013) to simulate the evolution of the snowpack (snow and
ice). A routine to account for the topographic shadowing effect of short wave radiation
(Revuelto *et al.*, 2016a) was included in the distributed simulations. The inclusion of
particular pixel features and topographic shadowing is the main difference between the
semi-distributed and distributed methods. Figure 2 shows a schematic representation of
distributed and semi-distributed simulation approaches.
3.3. Simulation initialization
Snowpack simulations were run for the period 1989–2015. However, the ISBA ground
state (including temperature and soil humidity) must be initialized to accurately
reproduce the evolution of the snowpack. A spin-up simulation for the 1988–89 snow
year (1 August 1988 to 31 July 1989) was repeated iteratively 10 times, to ensure a
realistic ground state when launching simulations.
Similarly, to adequately replicate the snow and ice evolution over glacierized areas a
glacier initialization was performed. Thus, for the simulations a sufficiently thick ice
layer (several tens of meters) was incorporated beneath the snow layers to ensure glacier
presence during each season in the glacierized areas. As Crocus is a multilayer
snowpack model that simulates the energy and mass interchanges between the various
snowpack layers, it also enables simulation of the glacier surface mass balance (Dumont
*et al.,* 2012a; Gerbaux *et al.*, 2005; Lejeune *et al.*, 2013). Glacierized areas were
initialized at the beginning of each snow season (1 August) using a 40-m ice thickness
(if the total ice thickness was less than this value), which ensured that it was present for
the entire snow season (from 1 August of one year to 31 July of the next year). Thus,
the six deepest Crocus layers were initialized with a density value of 917 kg/m$^3$ and a
temperature of 273.16 K (the Crocus default density and temperature values for ice, and
representative of temperate glaciers). The thickness of these layers progressively



transitioned from a shallow thickness for the upper layer (0.01 m) to thicker layers in
the deepest part of the ice (with a 5-fold difference factor between one layer and the one
above); this resulted in a total ice thickness of 39.06 m. The ice initialization was also
performed during the spin-up of soil to reproduce the ground state over glaciarized
areas. The extent of glacierized areas was based on the most recent data on their surface
area, inventoried in 2012 (Rabatel *et al.*, 2013). Although other historic surface
inventories of glacierized areas within the upper Arve catchment were available (1986
and 2003; Gardent *et al.*, 2014), the most recent inventory was used for simplicity
because the change in the glacierized surface area between the inventoried dates
represents less than a 1% of the total study surface area.
3.4 Evaluation strategy
The availability of direct snow and ice observations for mountain areas is limited.
Broadly, when the time between observations is short, the spatial extent is limited and
oppositely, when large areas are observed, the temporal frequency is low. Consequently,
evaluation of the performance of a model in reproducing the snowpack evolution is
difficult because of a lack of information. Although we did not evaluate a hydrological
model in this study, the "observation scale" defined by Blöschl and Sivapalan (1995)
aided assessment of the representativeness of the available observations. The
observation scale is defined by: i) the spatial/temporal extent (coverage) of a dataset; ii)
the spacing (space and time resolution) between samples; and iii) the integration volume
(time) of a sample (also known as support). These three criteria can rarely be optimized
simultaneously. Hanzer *et al.* (2016) introduced a representation to depict the suitability
of an observation dataset to evaluate model performance. To evaluate the simulations in
this study we used four datasets based on: *in situ* snow depth from Météo-France
stations; the snow covered area (SCA) from MODIS images; the punctual glacier
surface mass balance (SMB); and the glacier equilibrium-line altitude (ELA) from
Landsat/SPOT/ASTER. Based on the radar charts presented by Hanzer *et al.* (2016),
shown in their Figure 5, it was possible to fully evaluate the simulations using the four
observation datasets available for this study. The analyses presented below enabled us
to draw conclusions about the impact of the methods used on the various spatio-
temporal scales considered, also enabling an overall evaluation of the simulation
platform.





The four datasets used in evaluation of the simulations are described below. Not all
simulations (punctual, semi-distributed, and distributed) were evaluated using all four
observation datasets. The punctual snow depth simulations only provided a preliminary
evaluation of the simulation setup in terms of reproducing the temporal snowpack
evolution, so only punctual snow depth observations were used in the evaluation of this
simulation approach. The three other datasets (SCA, and glacier SMB and ELA) were
used in evaluating the semi-distributed and distributed simulations, as these datasets had
the appropriate spatial and temporal extents needed to assess the performance of these
two approaches.
*Punctual snow depth observations*
The Météo-France observation network has 5 stations in the study area (Fig. 1), located
at different elevations. Some of these stations acquired data during all snow seasons
throughout the entire study period, including at Nivose Aiguilles Rouges (2365 m.a.s.l.),
Chamonix (1025 m.a.s.l.), and Le Tour (1470 m.a.s.l.). Other stations were installed
later, and provided observational data since the 1994–95 snow season (Lognan station;
1970 m.a.s.l) and since the 2003–04 snow season (La Flegere station; 1850 m.a.s.l.). At
these stations the temporal evolution of the snow depth was observed at daily or sub-
daily time intervals, and these data were used to evaluate SAFRAN-Crocus in non-
glacierized areas during winter and spring (periods with snow presence).
*Snow cover area based on the MODIS sensor*
*i) Evolution of the snow covered area*
Many studies have demonstrated the usefulness of MODIS images for snow cover
mapping in mountain areas (Gascoin *et al.*, 2015; Klein and Barnett, 2003; Parajka and
Blöschl, 2008). The MODIS mission database provides long temporal coverage (the
mission was launched in 2000, and obtains daily images), so enabled a comparison
between the simulated and observed snow cover evolution for 14 snow seasons (out of
the 26) simulated on an almost daily basis (comparisons were limited by cloud cover in
the study area). Sub-pixel snow monitoring of the snow cover at 250-m spatial
resolution was performed using MODImLab software (Dumont *et al.*, 2012b; Sirguey *et*
*al.*, 2009). Multispectral fusion between MOD02HKM (500 m; bands 3–7) and
MOD02QKM (bands 1 and 2) (Sirguey *et al.*, 2008), enabled this software to generate
images at $250 \times 250$ m spatial resolution to derive various snow–ice products. We used
the unmixing_wholesnow (UWS) product, as it has been shown to outperform other





snow–ice products for assessing evolution of the SCA (Charrois *et al.*, 2013). We also
considered the cloudiness product in MODImLab to determine the proportion of the
catchment affected by cloud cover. Generation of the UWS and cloudiness products in
MODImLab software was based on the same DEM used for the snowpack simulations.
This ensured a direct match between of observation and simulation pixels. To avoid
errors related to cloud presence in the study area, only days having cloud cover
representing < 20% of the total surface area were considered in the analysis.
The UWS threshold for considering a pixel to be snow covered was set to 0.35 (i.e.,
fractional snow cover > 35%; Charrois *et al.*, 2013; Dedieu *et al.*, 2016). Three snow
depth threshold values (0.10, 0.15, and 0.20 m (Gascoin *et al*., 2015; Quéno *et al.*,
2016) were examined to consider a pixel as snow covered in the simulations.
The temporal evolution of the snow covered area (SCA) within the study area predicted
by each simulation approach (semi-distributed and distributed) was analyzed in terms of
the root mean squared error (RMSE), the mean absolute error (MAE), and $R^2$ for
comparisons between simulations and observations. The temporal evolution of the SCA
for specific snow seasons was also analyzed to assess the difference between
observations and simulations in different time periods. The SCA evolution in forested
areas was not evaluated, and these areas were masked in the analysis.
*ii) Evaluation of spatial similarity*
The spatial similarity between the observed and simulated SCA was evaluated for each
simulation approach based on two similarity metrics: the Jaccard index (J), and the
average symmetric surface distance (ASSD). As the grid cells coincided because the
simulations and observations were based on the same DEM, we were able to obtain
binary maps of snow presence from the simulated and observed maps, using the
thresholds established.
The Jaccard index is the ratio of the intersection between the observed (O) and the
simulated (S) SCA and the union of O and S (Equation 1). The index values range from
0 to 1, with a value of 1 representing a perfect match between the observed and
simulated SCA.
$$J = \frac{|O \cap S|}{|O \cup S|} \qquad (1)$$
The ASSD is complementary to J, as it evaluates the distance between the boundaries of
the observed and simulated SCA. ASSD is based in the modified directed Hausdroff
distance between boundaries (Dubuisson and Jain, 1994; see Quéno *et al.*, 2016 and





Sirguey *et al.*, 2009 for more details). The ASSD unit is meters, and the smaller the
distance the better the match between surface boundaries. The Jaccard index and ASSD
were calculated for the 2001–02 to the 2014–15 snow seasons. To assess the
performance of the two SCA simulation approaches for specific periods, the 2006–07
and 2007–08 snow seasons (both of which were characterized by low average levels of
snow accumulation) and the 2011–12 and 2012–13 snow seasons (characterized by high
levels of snow accumulation) were analyzed for both the accumulation period (January,
February, and March; JFM) and the melt period (May, June, and July; MJJ).
*Glacier surface mass balance*
Glaciers located in the Mer de Glace and Argentière sub-catchments have been
monitored, in a sufficient number of measurement locations for our analysis, since 1995
by the French Service National d'Observation GLACIOCLIM. During this period field
data were obtained twice per year, during the maximum (end April–May) and minimum
(around October) snow accumulation periods. These data enabled calculation of the
SMB for summer (SSMB; annual difference between the maximum and minimum
acquisitions), winter (WSMB; annual difference between the minimum of the previous
year and the maximum acquisitions), and annually (ASMB; year to year differences in
the minimum acquisitions) at each individual point of the network (Fig. 3). The
observation procedure involved use of glaciological methods (Cuffey and Paterson,
2010) to retrieve the surface mass balance for the various time periods (SSMB, WSMB,
and ASMB). Stakes (markers over the glaciers) are set up in both accumulation and
ablation areas throughout the glaciers, and so reflect the evolution of the various zones
of the glaciers. The spatial distribution of the stakes is shown in Figure 3. For further
information on the methods for SMB data collection, see Réveillet *et al.* (2017).
The observations of SMB for the various time periods at more than 65 locations
encompassing different glaciers enabled assessment of the snow and ice evolution over
glacierized areas, as these measurements included snow and ice ablation (SSMB) and
snow accumulation (WSMB) periods. Thus, the simulated SMB for the same
observation periods and locations were computed based on Crocus results. With this
information, a linear regression and $R^2$ coefficient were computed for each sub-basin for
the three periods, and these were used to measure the performance of the modeling
approaches. The simulated (distributed and semi-distributed) and observed temporal
evolutions of the SMBs were compared based on the SAFRAN elevation bands (the





average and standard deviation for all points within each band were calculated). To
assess any elevational dependence of the SMB, the seasonal evolution of the observed
and simulated SSMB, WSMB, and ASMB were compared for two snow seasons having
opposite characteristics (high and low levels of snow accumulation) for the Mer de
Glace glacier, which had a large gradient for assessing elevational dependence.
*Glacier equilibrium-line altitude*
The glacier equilibrium-line altitude (ELA) is the annual maximum elevation of the
snow–ice transition over glacierized areas. Since 1984 the temporal evolution of the
ELA for the five largest glaciers in the study area has been monitored using various
satellite sensors (Rabatel *et al.*, 2013, 2016). Data on the inter-annual evolution of the
ELA for the Tour, Argentière, and Mer de Glace glaciers (and its main tributaries, the
Leschaux and Talèfre glaciers) was available for the entire study period
Images from Landsat 4TM, 5TM, 7 ETM+, SPOT 1–5, and ASTER were used to obtain
the ELA for the study period. The spatial resolution of these images ranges from 2.5 to
30 m. The method of snow line delineation using multispectral images combining
green, near-infrared, and short-wave infrared bands has been fully described by Rabatel
*et al.* (2012). The satellite acquisition date depends on various factors including the
availability of satellite images for the study area and cloud presence, but images
obtained during the period of minimum snow accumulation (late August to early
October) were used to obtain the ELA. Thus, the simulated ELA was obtained for the
same dates as the satellite acquisitions. Because of the difference in the spatial
resolution of the simulation (250 m) and satellite observations ($\leq 30$m), the average and
standard deviations of the ELA were compared.





## 4. Results

### 4.1. Punctual snow depth

The observed and simulated snow depth evolution for the 2007–08 and 2012–13 snow seasons (low and high average snow accumulation years, respectively) for the five stations are shown in Figure 4. The snow depth evolution shows the capacity of the SAFRAN-Crocus model chain to reproduce the temporal evolution at locations having differing topographic characteristics.

It is important to note that the results shown in Figure 4 indicate the capacity of the simulations to reproduce snow depth dynamics at specific points having well known topographic characteristics. Punctual simulations include the impact of surrounding topography on incident solar radiation (terrain shadowing masks). Additionally, the meteorological forcing was taken at the station elevation (SAFRAN forcing not yet discretized on elevation bands). Nevertheless, the spatial scale of the meteorological forcing was that of the Mont Blanc SAFRAN massif. Therefore the spatial variability of solid/liquid precipitation within the massif is not taken into account.

Some snow accumulation events were underestimated or overestimated in the SAFRAN-Crocus simulation, evident in discrepancies between the simulated and observed snow depths, including for the Le Tour (overestimation) and La Flégère (underestimation) stations for the 2007–08 snow season. Despite these discrepancies resulting from meteorological forcing, the simulated evolution of the snow depth appeared reliable, in particular during melt periods.

Table 1 shows the RMSE and bias errors between observations and simulations at the five stations. There was a high level of variability between the errors for the various stations, mainly because all local effects were not included in the simulations. It is noteworthy that the number of observations available and the time periods (which could have marked differences on total seasonal snow accumulation) affected the significance of the RMSE and bias for the various stations (Table 1). The RMSE values ranged from 20.8 to 66.6 cm and the bias ranged from –19.1 to 49.4 cm. These values are small relative to the total snowpack thickness (snow depth observations were commonly > 200 cm, and in some cases exceeded 300 cm). However, for the Aiguilles Rouges station the RMSE and bias estimates were higher than for the other stations. This may be because this station is exposed to major wind-induced snow transport episodes that were not accounted for in the simulation. In addition to these events, this station is also



affected by forecasting errors related to the meteorological forcing, such as the large
underestimation for the first snowfall in 2007–08.
4.2. Snow Cover Area evaluation
Figure 5 shows an example of the SCA obtained using the UWS product for 24 July
2008, and the corresponding simulated snow depth determined using the distributed
approach. This date was selected because it was a cloud-free day with high elevation
areas covered by snow.
Table 2 shows the SCA simulation results estimated based on 0.1, 0.15 and 0.2 m snow
depth thresholds compared with the observed UWS (0.35 threshold), for the 2008–09
and 2009–10 snow seasons (average snow accumulations). In light of these results we
selected a 0.15 m snow depth simulation threshold for deciding whether a pixel was
snow covered.
*i) Evolution of the snow covered area*
The results of simulation of the SCA in the study area for 10 of the 14 snow seasons
(for ease of visualization) based on MODIS data are shown in Figure 6. This figure
shows that both approaches were able to reproduce the SCA evolution based on MODIS
images. During winter and early spring, when large areas of the catchment are covered
with snow, there was a high degree of consistency between the observations, and
simulations based on each approach. In contrast, during summer and early autumn,
when snow is only present at high elevations and on preferential accumulation areas,
there was less consistency between observations and simulations, particularly for the
semi-distributed simulations.
Figure 7 shows the SCA evolution for four non-consecutive snow seasons, two having
low levels of snow accumulation (2006–07 and 2007–08 seasons) and two having high
levels of snow accumulation (2011–12 and 2012–13 seasons). In winter the simulation
slightly overestimated the SCA compared with observations, but during summer and
autumn the simulations underestimated the SCA. However, the distributed simulations
most closely reproduced the observed SCA (Table 3). In all four seasons the semi-
distributed simulations generated larger underestimates of the SCA during summer and
early autumn.
Using the terrain aspect classification for semi-distributed simulations it is possible to
evaluate the impact of terrain shadowing effects. From the eight orientation classes we
identified two main groups: those having a northern aspect (N, NW, NE) and those



having a southern aspect (S, SE, SW). Figure 8 shows the observed and simulated SCA
evolution for high and low snow accumulation seasons in relation to these two terrain
classes. The variability in the SCA was well captured for both aspects by both the semi-
distributed and distributed simulations. Error estimates for the SCA simulated in
relation to the north and south aspects (Tables 4 and 5) were lower for the distributed
simulations compared with the satellite observations. Moreover, the SCA temporal
evolution shown in Figure 8 shows that overall the simulation underestimated the SCA,
during late spring and summer in northern aspects. For southern aspects, simulation of
the SCA evolution was poorer during winter.
*ii) Evaluation of the spatial similarity*
The spatial similarity between the observed and simulated SCA is exemplified in the
temporal evolution of the Jaccard index and ASSD. Table 6 shows the average values
for J and ASSD for the entire study period and for the 2006–07 and 2007–08 snow
seasons (low levels of snow accumulation) and the 2011–12 and 2012–13 snow seasons
(high levels of snow accumulation).
The higher scores found during seasons having high levels of snow accumulation were
expected because of the larger areas covered by snow. Figure 9 shows the temporal
evolution of the Jaccard index and ASSD for high and low level snow accumulation
seasons. Although the difference between the distributed and semi-distributed
simulations was low for most dates, the Jaccard index values for the distributed
simulations were higher, showing a greater capacity for simulating the SCA (Table 6).
Similarly, ASSD values were lower for distributed simulations, which showed reduced
distances between the limits of snow free and snow covered areas. The differences
between the two approaches are also evident in the average values shown in Table 6.
The performance of the simulations appeared to differ between periods of maximum
and minimum snow accumulation (Fig. 9). Table 7 shows the average Jaccard and
ASSD index values obtained for the JFM and MJJ periods for the four snow seasons
analyzed in detail (high and low level snow accumulation seasons). The better
performance of distributed simulations was a result of better reproduction of the SCA
evolution, and their ability to capture better spatial patterns in heterogeneous mountain
terrain. Not surprisingly, the values in Table 7 also show higher scores for both
simulations during winter and early spring, when the SCA was high.





4.3. Glacier surface mass balance
Analysis of the glacier surface mass balance enabled assessment of the effectiveness of
simulations of the seasonal and annual evolution of snow and ice on glacier surfaces.
Figures 10 and 11 show the simulated and observed temporal evolution of the surface
mass balance for the 300-m elevation bands. These show good agreement between
observations and simulations with respect to year-to-year SMB variability. During
winter the snow accumulation at high elevations was underestimated. For elevations
above 2700 m.a.s.l. a higher positive glacier SMB was observed, and the difference
between the observed and simulated SMB increased at higher elevations. During
summer, when solid precipitation has no or marginal influence in low elevation areas
and little influence at higher elevations, the observed and simulated SMB values were
similar for elevations above 2100 m.a.s.l. for the Mer de Glace glacier, and above 2400
m.a.s.l. for the Argentière glacier. Nevertheless, in high elevation areas the SSMB
deviation was also underestimated on the simulations. This was probably because of the
lower level of snow accumulation simulated during winter (using SAFRAN model)
which induces an earlier complete melting of snow in the simulation in low elevations.
This is presumably because of more rapid melting of ice insulated from solar radiation
by the snow layers above, and because of the impact of variations in wind speed or long
wave radiation on the simulation.
Combination of the simulated WSMB and SSMB produced an ASMB that
underestimated snow accumulation at high elevations (> 3000 m.a.s.l.) and melting at
low elevations (2400 m.a.s.l. for the Argentière glacier, and < 2400 m.a.s.l. for the Mer
de Glace glacier). Thus, the glacier ASMB included summer and winter variations,
which in some cases negated each other. The contrasting performance of the simulations
in reproducing the SMB between high and low elevations is clearly illustrated in Figure
12. This shows the altitudinal dependence of the SMB for two snow seasons, one
having a low level of snow accumulation and the other a high level. The simulated
SSMB, WSMB, and ASMB values for both approaches underestimated the observed
values at both low (higher negative loss of water equivalents observed) and high (lower
positive loss of water equivalents observed) elevation areas. Nevertheless, the SMB
simulations at intermediate elevations correctly reproduce the observed values, and the
temporal evolution of the SMB for the 20 years (Figs 10 and 11) was well reproduced
by the simulations.



The performance of simulations in reproducing glacier SMB must take account of the
areal extent at differing elevations.  Elevations > 3000 m.a.s.l. represent 37% and 52%
of the surface areas of the Argentière and Mer de Glace glaciers, respectively. The
Argentière glacier has < 10% of its surface area below 2400 m.a.s.l., and the Mer de
Glace glacier has < 7% below 2100 m.a.s.l. These relative extents of glacierized surface
area show that for large areas of the glaciers the SMB was accurately reproduced by the
simulations. However, for large glacierized areas there were marked differences
between the observations and simulations; although the year-to-year evolution was
accurately reproduced, this demonstrates the need to improve simulation methods.
In general, the distributed simulation values for the SMB were slightly closer to the
observed SMB values than were those from the semi-distributed simulations. Table 8
shows that the RMSE values were lower for the distributed simulations and the $R^2$
values were higher for most periods in both glacierized areas. However, the WSMB
simulations obtained using the semi-distributed approach were slightly better at
reproducing the SMB.
4.4. Glacier Equilibrium Line Altitude
The temporal evolution of the ELA for the five largest glaciers in the study area is
shown in Figure 13. Overall, and despite differences in the spatial resolutions of
simulations and observations of ELA, the ability of the simulations to capture the
temporal evolution of the ELA during the 26 years of the study was satisfactory, with
lower variations found for distributed simulations for most seasons.
Table 9 shows the average absolute differences between observations and simulations
and the linear adjustments for the five glaciers. These results show a systematic positive
bias on the simulated ELA which is consistent with the summer underestimation
revealed by the previous tests.





**5. Discussion**

5.1. Overview of SAFRAN-Crocus performance

The observation dataset used in this study enabled multilevel spatio-temporal validation of the performance of snowpack simulations at the scale of a large alpine catchment. The analysis of the results of semi-distributed and distributed simulations provided a holistic evaluation of the snow and ice dynamics in the study area. Overall, the SAFRAN-Crocus simulations have shown a good capability on reproducing the temporal evolution and spatial variability of snow and ice during the study period.

The simulations were evaluated using snow depth data from five Météo-France stations. Their ability to reproduce a bulk variable such as snow depth suggests that the main simulation processes were satisfactory, especially those related to the various components of the energy and mass balance. These findings are consistent with previous evaluations of the SAFRAN-Crocus system (Durand *et al.*, 2009a; Lafaysse *et al.*, 2013).

Distributed information on the snowpack evolution from the MODIS sensor enabled evaluation of the simulation results on a suitable temporal scale. Although many MODIS images were discarded because of cloud cover, they demonstrated the capacity of SAFRAN-Crocus to simulate the spatial distribution of the SCA over time for large areas having high spatial heterogeneity. The 14-year time period spanned is longer than in all previous similar evaluations, and at a higher spatial resolution (Quéno *et al.*, 2016). Evaluation of the spatial similarity between simulations and observations (Jaccard index and ASSD) showed that the SCA spatial pattern was well reproduced. The simulated SCA for winter was in close agreement with observations, as most of the study area was covered by snow. In contrast, during summer the performance of simulations declined, as evidenced by the increase in ASSD and the decrease in the Jaccard index. As small scale topographic effects that control snow accumulation on preferential accumulation areas were not included in the simulations, deviations from observations would have increased for certain periods, particularly the late melt period. These processes, which are mainly driven by small topographic features, can be long-lasting during the late melt period (Revuelto *et al.*, 2016b; Sturm and Wagner, 2010). This was particularity evident in comparisons of the scores for the 2006–07 and 2007–08 periods with those for the 2011–12 and 2012–13 periods (Table 3). The differences in response may have originated from the higher weight of glacier melt processes in



years with shallow snow depth. For these years, the good capability of the model on
reproducing snow melting is lumped because the snow distribution is not appropriately
simulated.
The availability of observations of the glacier SMB over a long time period provided an
opportunity to evaluate the performance of the simulations in capturing the snow and
ice temporal evolution over a wide range of elevations over glacierized areas.
Contrasting simulation performances were found in the various elevation bands, and
changed with the time period involved (summer, winter, or annual scales). The
performances in simulating the SMB for the Argentière and Mer de Glace glaciers
differed at high and low elevations. Although the observed SMB was always higher
than the simulated one for elevations exceeding 2700 m, the opposite was observed for
areas having elevations below 2100–2400 m. As the temporal variability of solid
precipitation generally explains the temporal variability of the WSMB (Réveillet *et al.*,
2017), it is important to consider differences between simulated and observed solid
precipitation, and how these could affect underestimation of the SMB in simulations.
Studies in the same study area and nearby glaciers suggest that at high elevations the
SAFRAN reanalysis may underestimate solid precipitation at ratios ranging from 1:1.2
at 2000 m.a.s.l. and 1:2.0 at 3200 m.a.s.l., with an average of 1:1.5 at the glacier scale (
Gerbaux *et al.*, 2005; Réveillet *et al.*, 2017; Viani *et al.*, submitted). This mainly results
from the lack of precipitation observations at high elevations available for assimilation
into the SAFRAN reanalysis; consequently divergences increase with elevation. Despite
this shortcoming, the simulations captured the inter-annual fluctuation of the WSMB for
all elevation bands. During summer the SMB could be explained by temperature
variability in the two glaciers (Réveillet *et al.*, 2017), thus simulations results are closer
to observations, particularly at higher elevations. In summer, most precipitation is
liquid, and so has little impact on the energy balance of the glaciers (Hock, 2005); this
may explain the improvement in summer simulations for most elevations.
It has recently been shown that Crocus is able to accurately simulate snow albedo
(Réveillet *et al.*, in prep), which is important because of its influence on the surface
mass balance (Essery and Etchevers, 2004; Essery *et al.*, 1999). However, it has been
demonstrated that Crocus results are directly affected by uncertainties in the estimation
of long wave radiation and wind (Réveillet *et al.*, in prep). Such effects may be
significant for elevations where the snow completely melts during summer and do not



insulate ice from the atmosphere during late melt season; this includes the low elevation
areas of glaciers, where high SSMB errors were found. At the annual time scale, glacier
differences between the observed and simulated SMB at high elevations during winter
and at low elevations during summer were reduced because the SMB underestimates for
winter (note these were negative/positive at high/low elevations) were compensated for
by more accurate simulations during summer, and vice versa. Regardless of these errors,
SAFRAN-Crocus was able to replicate the interannual evolution of the SMB.
Additionally, there was a good match between observations and simulations for the
2100–2400 to 3000 m.a.s.l. elevation bands for the Mer de Glace and Argentière
glaciers, respectively; these elevation bands encompassed large proportions of the
glaciers (approximately 40 and 53%, respectively).
For the entire study period the SAFRAN-Crocus simulations effectively reproduced the
observed inter-annual evolution of the study area glacier ELA. However, some
differences were evident, particularly on steeper glaciers, because the high spatial
heterogeneity was not well captured by the simulations. For mid-latitude mountain
glaciers, the annual evolution of the ELA can be considered to be a good proxy for the
glacier surface mass balance (Braithwaite, 1984; Rabatel *et al.*, 2005). Thus,
observations of the glacier SMB, together with the ELA, provide for a complete
evaluation of glacier temporal evolution.
5.2. Limitations of the evaluations performed
Although the observation dataset enabled comprehensive evaluation of the simulations,
it had limitations. First, the discrepancy in spatial scale between the SAFRAN
meteorological analysis and the snow depth observations, and the low number of
stations, limited the interpretation of results in terms of the simulated snow depth.
Differences in the temporal evolution of snow depth between observation and
simulations were in part associated with the unresolved sub-massif spatial variability in
the level of precipitation, as previously described (Durand *et al.*, 2009a; Lafaysse *et al.*,
2013; Vionnet *et al.*, 2016). *In situ* observations are also subject to local effects
associated with the topographic control at each site, including exposure to dominant
winds, which markedly affects the snow depth dynamics. Such effects remain difficult
to capture in snowpack modeling (Dadic *et al.*, 2010a; Liston *et al.*, 2007; Revuelto *et
al.*, 2016a; Schirmer *et al.*, 2011; Vionnet *et al.*, 2014), and were not included in the
modeling involved in our study. Discrepancies originating from the snow–rain limit can



also influence the snow depth. Stations at high elevation (Aiguilles Rouges: 2365 m.a.s.l.) are typically not affected by this phenomenon during winter, as the 0°C isotherm is located at lower elevations. In contrast, low elevation stations (Le Tour: 1470 m.a.s.l.; Chamonix: 1025 m.a.s.l.) are potentially affected by differences between the simulated and observed snow–rain limit, even during winter. In mid-latitude regions including the Alps, elevational shifts in the 0°C isotherm cover a significant variation throughout the year, including the elevations where each of the stations in this study is located.

Data on the spatial extent of SCA derived from MODIS images enabled distributed evaluation of the simulations. However, its usefulness in analysis of the performance of spatial simulations is limited, as it does not provide information on other snowpack variables, and imposes restrictions on the spatial resolution. Satellite observations also involve uncertainty, depending on the routines applied for generating the final product and the thresholds used to decide whether a pixel area as covered by snow. We adopted a 0.35 UWS threshold for considering a pixelas snow covered in satellite imagery (Charrois *et al.*, 2013; Dedieu *et al.*, 2016). We also performed an analysis to select the simulated snow depth threshold for considering a pixel to be snow covered. The 0.15 m threshold selected is consistent with values reported in previous studies (Gascoin *et al.*, 2015; Quéno *et al.*, 2016). In addition to the above issues, satellite products can have errors for specific dates. For a small number of days during the study period the SCA obtained from MODIS images did not describe the real extent of snow cover. For these days the SCA did not match the temporal SCA evolution observed on previous and later dates. Furthermore, days having the maximum cloud cover allowed in our analysis could have ± 20% SCA variability. This induces uncertainty in the observation for certain dates which can be greater than this of the pixel classification as snow covered in the simulations (note the ± 0.05 m snow depths threshold tested). In addition, pixels classified as snow covered in which bare soil may have a non-negligible extension (pixels close to the 0.35 UWS threshold) could introduce discrepancies between observations and simulations, mainly during summer.

Glacier surface mass balance observations also involve limitations. For instance, infrequent glacier SMB observations for certain temporal windows limited evaluation of the simulated SMB. The spatial sampling involved in the glaciological method can also be a significant source of uncertainty, especially for elevation bands for which there are




a limited number of observations. Additionally, the average SMB obtained for the
elevation bands can lump the high SMB spatial variability that occurs within a specific
band. For most years and all the elevation bands the uncertainty associated with the
average SMB measurements (± 0.2 m water equivalent; Réveillet *et al.*, 2017) was
exceeded by the uncertainty associated with the observations for each band. This could
have affected the results presented here, indicating that the standard deviations for the
observed SMB values should be retained when analyzing the results of the simulations.
The simulations underestimated the observed SMB for the lowest elevations having
SMB observations, despite the temporal variability being replicated. This may have
been related to errors in precipitation and phase, and in this regard differences in the
snow–rain limit could be important. Additionally, the impact of local effects is more
important at low elevations, as glaciers are more confined in valleys that have very
steep slopes and adjacent high mountains. In low elevation areas, where ice is exposed
to the atmosphere for longer periods during the year (snow does not insulate ice from
the atmosphere since it has disappeared), differences in meteorological forcing variables
including wind and temperature can have a marked influence on simulation results
(Réveillet *et al.*, submitted). Similarly, at low elevations the glaciers are usually covered
by debris, as is the case for the Mer de Glace glacier. This was not considered in our
simulations, but differences in the behavior of the snow–ice interface in debris-covered
areas could be expected to affect the simulation results (Lejeune *et al.*, 2013).
Some issues were also evident in evaluation of the ELA. For the smallest glaciers, a
reduced number of pixels having the 250-m pixel resolution were considered. As the
ELA observations were based on Landsat, SPOT and ASTER satellite images (2.5–30
m resolution) the spatial variability of the simulation made it difficult to identify the
glacier margins. The combination of problems in delimitating glaciated areas over
smaller ice bodies, and the smooth topography characterizing the simulations compared
with real terrain, could cause simulation errors for smaller glaciers.
5.3. Distributed *vs.* semi-distributed approaches
In this study we performed distributed and semi-distributed snowpack simulations using
the same model and evaluation setup (including ice initialization, meteorological
forcing, projection on the same grid, observation databases). Thus, both approaches
were affected by the same methodological limitations. The simulation results were
consistent with the observed SCA evolution using both approaches. However, better



results were obtained from the distributed simulations, especially during late summer.
The energy balance was more accurately simulated in the distributed approach, as it
accounted for terrain shadowing effects on incoming solar radiation. The distributed
simulations also accounted for the specific characteristics of each pixel rather than
categorization based on topographic classes. The distributed approach also produced
more accurate simulations of the SCA for the various time periods, particularly during
the late melt period. Similarly, spatial similarity evaluation (Jaccard index and ASSD)
also showed that the distributed approach was slightly superior at reproducing the SCA
distribution. The semi-distributed approach better simulated the temporal evolution of
the SCA for areas having a southern aspect, because of terrain shadowing effects in
areas having a northern aspect are not appropriately considered. Oppositely, the
simulation in northern aspects obtained with the distributed approach is superior
because these are able to include terrain shadowing on the simulations.
Based on the glacier SMB scores and their temporal evolution, we concluded that the
best simulation approach depends on the season involved. Thus, the WSMB evaluation
showed that similar results were obtained using the two methods. In contrast, the
distributed approach was better at simulating the SSMB. The similar performances of
the semi-distributed and distributed simulations during winter, but the better results for
the distributed simulations for summer resulted in the distributed approach providing
greater accuracy at the annual scale. The better results obtained for both glaciers
analyzed for a long time period (ASMB) using the distributed simulations suggests that
this approach is likely to provide more reliable results over longer periods.
The distributed simulation of the ELA generally showed closest agreement with
observations, but for certain years the semi-distributed simulations most accurately
reproduced the observed values. Thus, it is not possible to conclude that one approach
to reproducing the ELA was superior. This uncertainty may be related to the coarse
pixel size, which did not enable the high spatial heterogeneity of the terrain to be
captured. The annual ELA covers a small area of the glaciers (it represents the snow line
limit between snow-free and snow-covered areas), and thus the effect of spatial
heterogeneity is likely to be significant.
Overall, the distributed simulations were better at reproducing observational data. Thus,
distributed simulations, which better represent the spatial heterogeneity of mountain
areas, in general produce more accurate snowpack simulations, and are the





recommended modeling approach. However, depending on the purpose of the
simulations and the accuracy required, other factors must be considered. For instance,
semi-distributed simulations have lower computing resource requirements; in this study,
the distributed approach had computing requirements that were a factor of 100 greater.
The accuracy of semi-distributed simulations in reproducing the snowpack evolution
over large areas makes them useful in many applications.
5.4. Future perspectives on distributed snowpack simulations
Simulating the snowpack evolution in mountain areas is challenging. Although
advances in meteorological/snowpack models and simulation approaches are improving
the reproduction of observational data, inaccuracies remain. Many studies have
highlighted the potential to improve snowpack modeling by assimilating observational
data (Griessinger *et al.*, 2016; Thirel *et al.*, 2013). Satellite data enables the distribution
of the snowpack over large areas to be determined, and the assimilation of such data
into snowpack models has been shown to significantly improve the simulation results
(Charrois *et al.*, 2016). In distributed snowpack simulations almost direct satellite data
can be assimilated, in contrast to the semi-distributed approach. Additionally,
meteorological forcing models having high spatial resolution are improving simulations
of the spatial pattern of meteorological variables in mountain areas (Schirmer and
Jamieson, 2015; Vionnet *et al.*, 2016; Weusthoff *et al.*, 2010). This will improve
snowpack simulations (Förster *et al.*, 2014; Quéno *et al.*, 2016), even though it is
challenging to combine high resolution numerical weather prediction models with
precipitation measurements assimilation in analysis systems. Interest in distributed
snowpack simulations will be enhanced when reliable high spatial resolution
meteorological forcing data are available, as only this simulation approach can take full
advantage of such data. Further research is needed on parameterizing small scale
snowpack processes for incorporation in modeling, including wind driven snow
transport (Dadic *et al.*, 2010b; Winstral *et al.*, 2012), avalanche snow redistribution
(Bernhardt and Schulz, 2010), and topographic control on snow distribution (Revuelto
*et al.*,. 2016a). Inclusion of these processes, together with the incorporation of reliable
meteorological forcing and satellite data, assimilation will improve the accuracy of
snowpack simulations over extensive mountain areas.



**6. Conclusions**

This study provided a detailed assessment of the ability of the SAFRAN-Crocus system
to simulate the snow and ice dynamics in complex alpine terrain using distributed and
semi-distributed simulation approaches. The study was undertaken in the upper Arve
catchment in the western French Alps, with simulations run for the 1989–90 to the
2014–15 snow seasons.

A preliminary evaluation of the simulations was completed based on observations of
snow depth derived from five meteorological stations within the study area. This was
only performed using punctual snowpack simulations, to provide an initial assessment
of model performance over non-glaciated terrain. Despite some discrepancies between
observations and simulations, the model reliably reproduced the snow depth, especially
during melt periods.
In regard to the spatial scale of snowpack simulations over extended areas, the semi-
distributed and distributed simulations were compared using the same observation
datasets, including: (i) the temporal evolution of the snow-covered area based on data
from the MODIS sensor; (ii) measurements of surface mass balance of glaciers within
the upper Arve catchment; and (iii) observational data on the annual evolution of the
equilibrium-line altitude for the various glaciers considered.
Both simulation methods accurately reproduced the evolution of the SCA during
accumulation events, as they relied on the same meteorological forcing data. For the
winter to early spring period, when the study area is almost completely covered by
snow, there was little difference between the two approaches. However, for the melt
period the distributed simulations better reproduced the observations.
The simulations for low elevations and elevations > 2700 m.a.s.l. underestimated
(negative underestimation in low elevations and positive in high) the observed SMB.
Nevertheless, the results of both simulations were in close agreement with observations
at mid-elevation areas, and adequately reproduced the observed annual SMB at all
elevations. Overall, the distributed simulations yielded better results.
Based on comparison with ELA data obtained from various satellites at the end of
summer, the SAFRAN-Crocus accurately reproduced the inter-annual variability of the
snowpack over glaciated areas. However, differences between observations and
simulations were evident, particularly for the smallest glacierized areas, where the
spatial resolution of the simulations did not enable the high spatial variability of the



topography to be included. In addition, based on the ELA evaluation, the distributed
approach was slightly better at reproducing the snowpack dynamics.
Overall, the results of this study demonstrated that distributed simulations were better at
reproducing snowpack dynamics in the alpine terrain of our study area. Distributed
simulations take account of the specific topographic characteristics of each pixel and
also the effects of terrain shadowing by surrounding areas. Inclusion of these two effects
over long time periods led to better results being obtained using the distributed
approach. Distributed simulations will facilitate incorporation of the latest snowpack
modeling advances, including assimilation of satellite data and the use of higher spatial
resolution meteorological forcing models.





**7. Acknowledgments**
This study was funded by Syndicat mixte d'aménagement de l'Arve et de
ses abords (SM3A), Communauté de Communes de la Vallée de Chamonix Mont-Blanc
and Fondation Terre Solidaire in the framework of the Programme
d'Action de Prévention des Inondations (PAPI). We thank Glacioclim
(https://glacioclim.osug.fr) for generating the glacier surface mass balance database
used in the study. J. Revuelto benefited from a grant within the above-cited PAPI
project and is now supported by a Post-doctoral Fellowship of the AXA research found
(le Post-Doctorant Jesús Revuelto est bénéficiaire d'une bourse postdoctorale du Fonds
AXA pour la Recherchem Ref: CNRM 3.2.01/17). IGE and CNRM/CEN are part of
Labex OSUG@2020.



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

plant responses to changes in snow depth and snowmelt timing. Clim. Change *94*, 105–

1155   121.





**Figures**

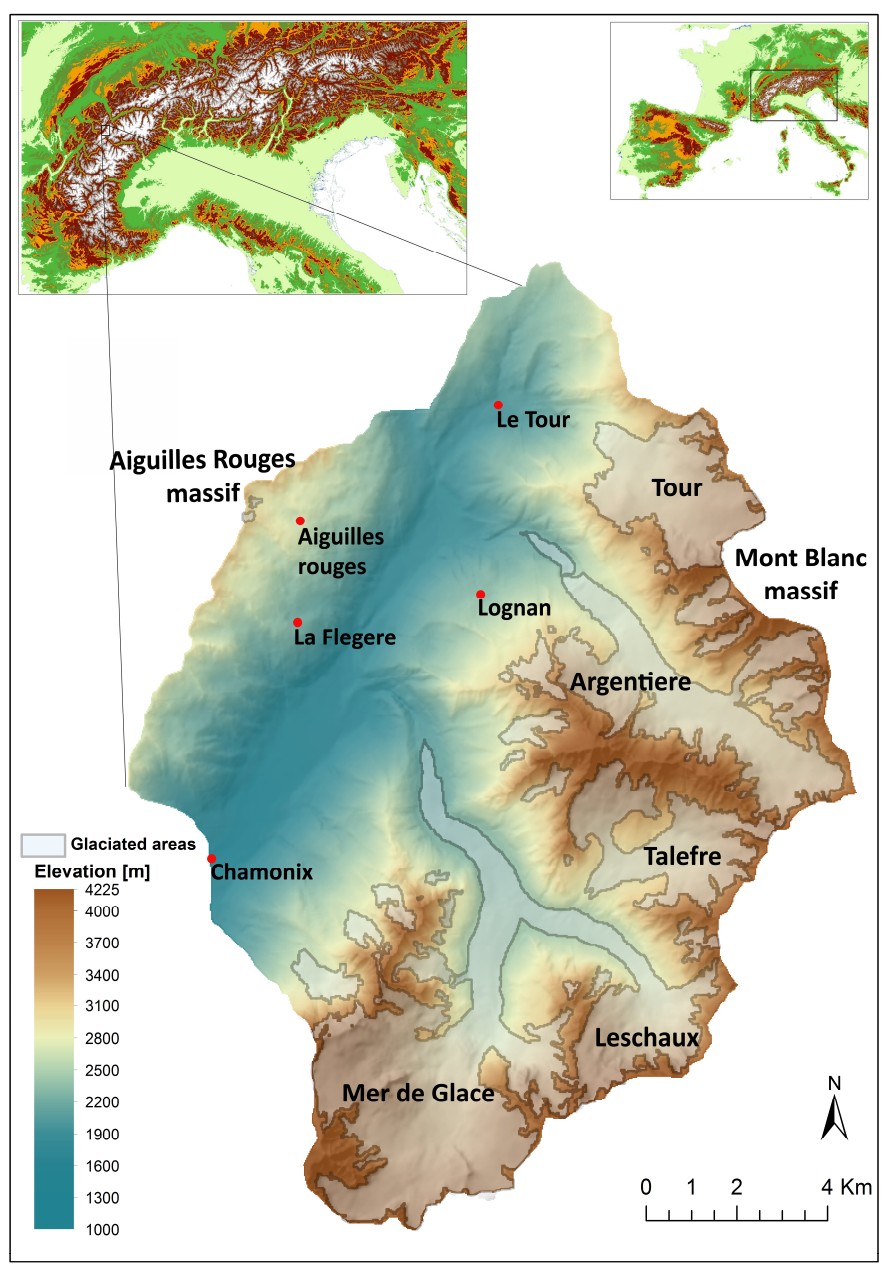

**Figure 1:** Upper Arve catchment study area. The white shaded area shows the extent of
the glaciers in 2012 (Gardent *et al.*, 2014). The inner maps show various magnifications
of the Alps and the location of the Arve valley within the mountain range. The red
points show the position of the five Météo-France stations located in the study area.






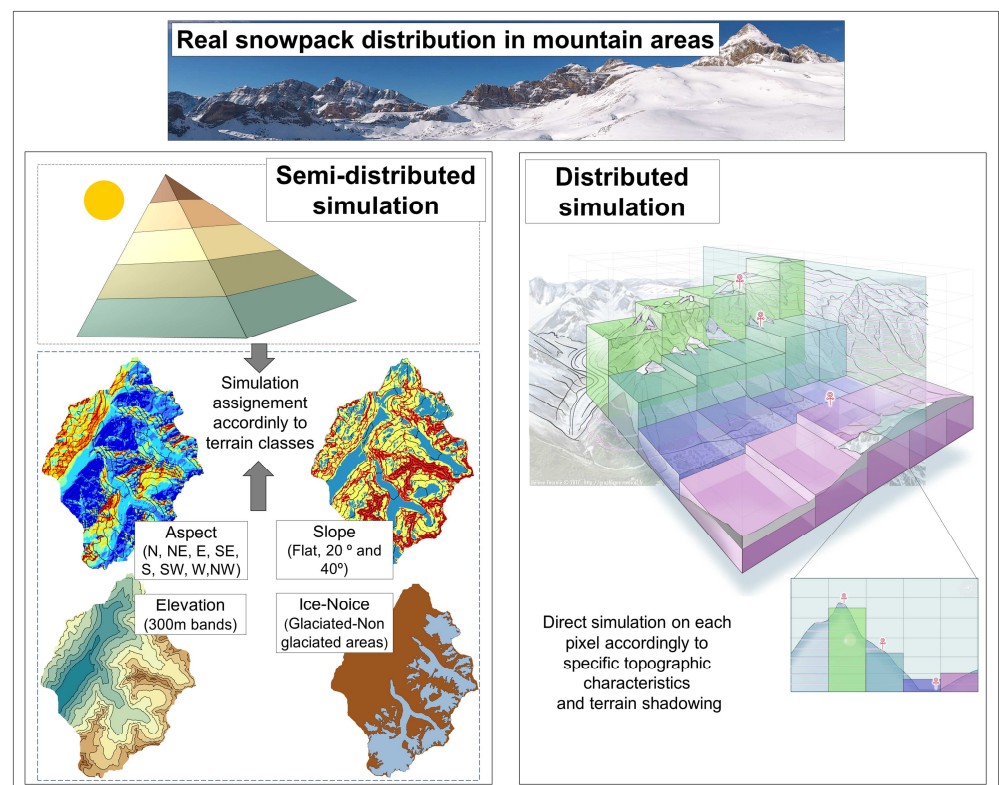

**Figure 2:** Schematic representation of the approaches used to account for mountain
spatial heterogeneity when simulating snowpack dynamics.

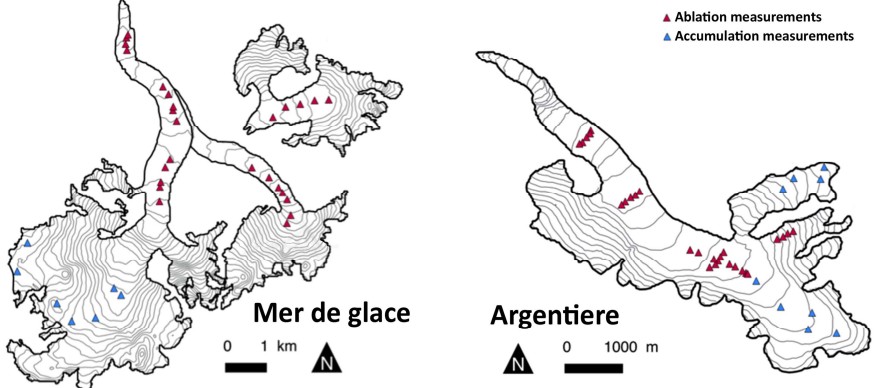

**Figure 3:** Glacier SMB measurement locations for ablation and accumulation areas in
the Mer de Glace and Argentière glaciers.

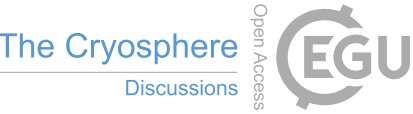

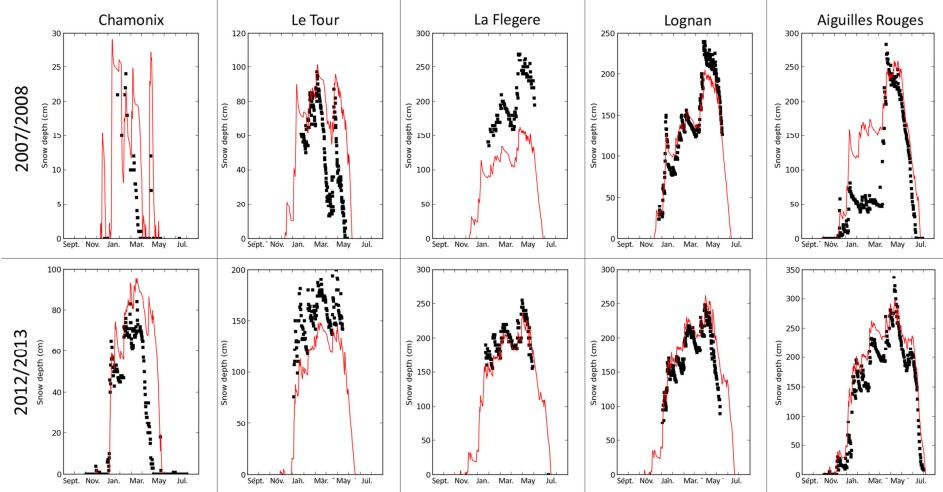

**Figure 4:** Observed (black squares) and simulated (red lines) snow depth evolution for the 2007–08 (upper panel) and 2012–13 (bottom panel) snow seasons. The elevations of the stations are: Chamonix: 1025 m.a.s.l.; Le Tour: 1470 m.a.s.l.; La Flegere: 1850 m.a.s.l.; Lognan: 1970 m.a.s.l.; and Aiguilles Rouges: 2365 m.a.s.l.

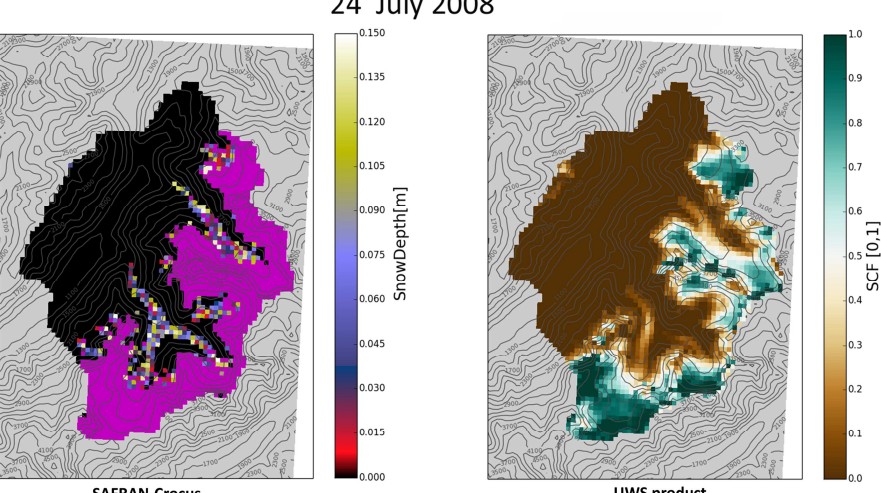

**Figure 5**: Spatial distribution of the UWS MODImLab product (equivalent to the SCA distribution), and the simulated snow depth obtained using the distributed approach (the purple color shows the snow depth values exceeding the 0.15 m threshold) for 24 July 2008.



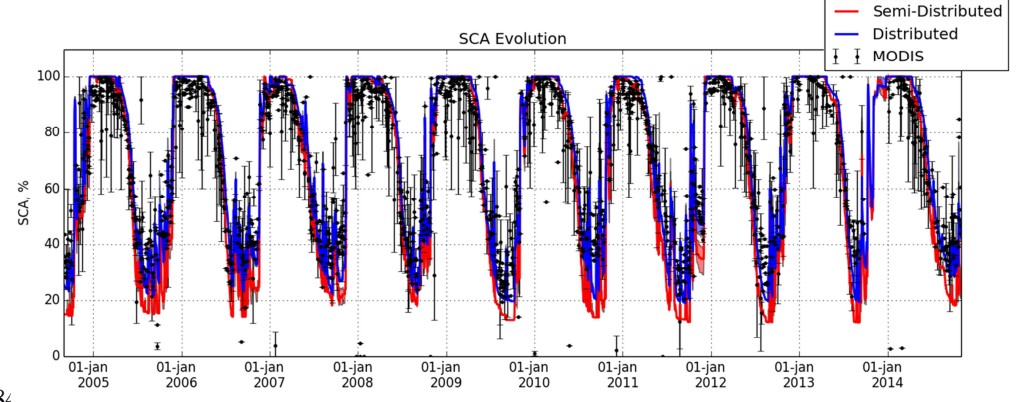

**Figure 6:** Temporal evolution of the SCA (2004–2014) based on semi-distributed and distributed simulations and MODIS sensor observations. The vertical bars associated with the MODIS observations show the uncertainty associated with cloud presence for days having ≤ 20% snow cover.





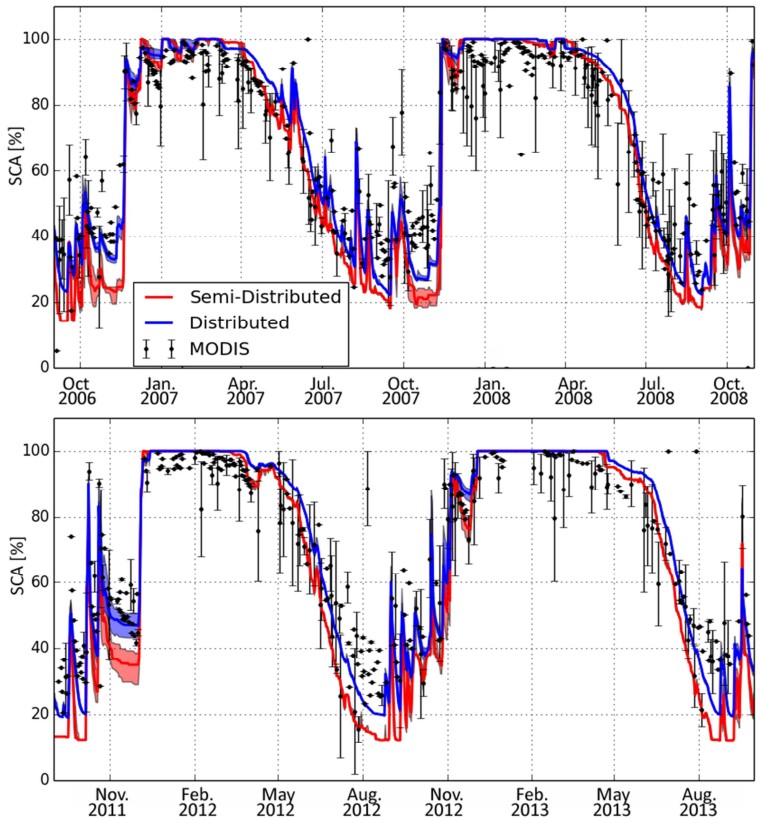

1190

**Figure 7:** Observed and simulated SCA evolution for a period of low level snowpack
accumulation (2006–2008; upper panel) and a period of high level snowpack
accumulation (2011–2013 lower panel). The vertical bars for the MODIS observations
show the uncertainty associated with cloud presence for days having ≤ 20% snow cover.
Red and blue shading for the distributed and semi-distributed SCA simulations show the
uncertainty associated with various snow depth thresholds for determining whether a
pixel was snow covered. The lower limit of the shading represents the SCA evolution
for a 0.1 m threshold, the upper limit of the shading represents a 0.2 m snow depth
threshold, and the middle line represents a 0.15 m snow depth threshold.

1200



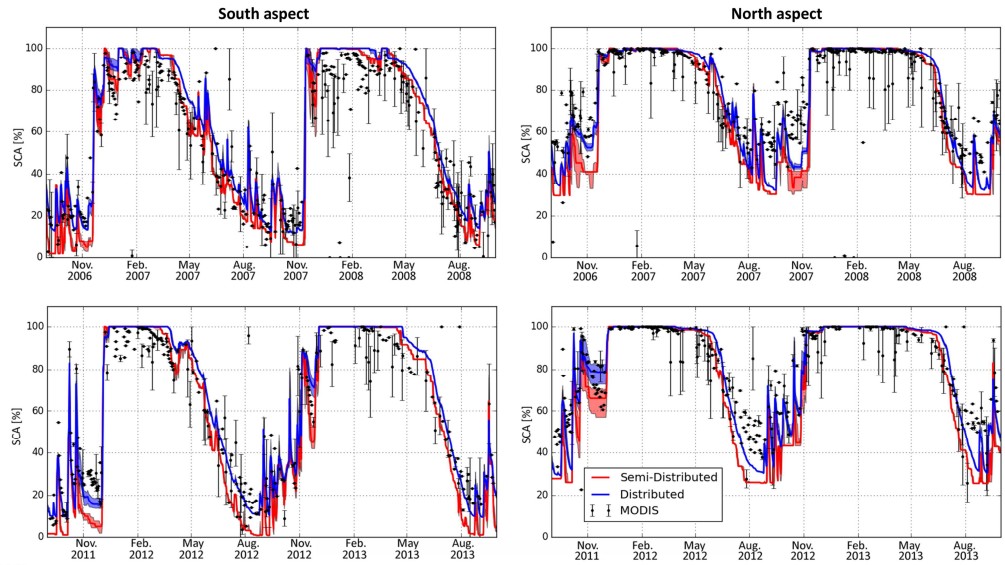

**Figure 8:** Evolution of the SCA in relation to north and south aspect for the 2006–2008 (upper panel; low level of snowpack accumulation) and 2011–2013 (lower panel; high level of snowpack accumulation) snow seasons. Vertical bars for the MODIS observations show the uncertainty associated with cloud presence for days having ≤ 20% snow cover. Red and blue shading for the distributed and semi-distributed SCA simulations show the uncertainty associated with various snow depth thresholds for determining whether a pixel was snow covered. The lower limit of the shading represents the SCA evolution for a 0.1 m threshold, the upper limit of the shading represents a 0.2 m snow depth threshold, and the middle line represents a 0.15 m snow depth threshold.

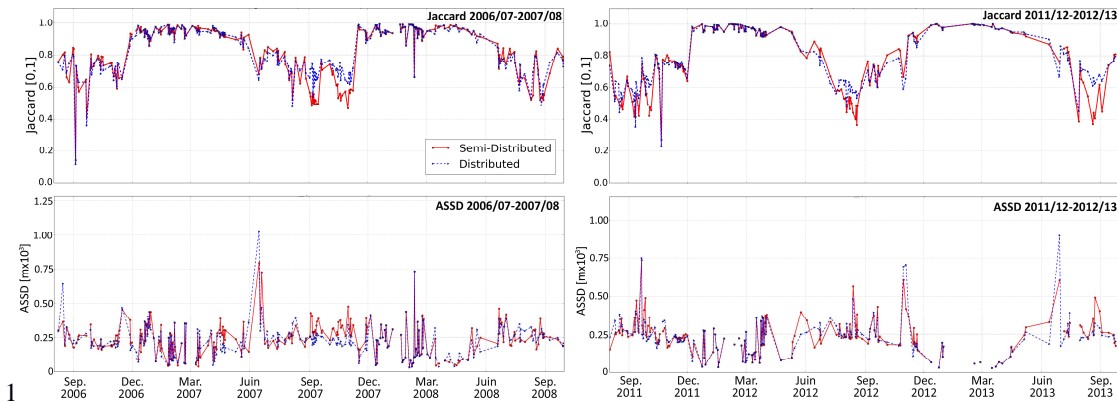

**Figure 9:** Jaccard index and ASSD values for low level (2006–07 and 2007–08) and high level (2011–12 and 2012–13) snow accumulation seasons.




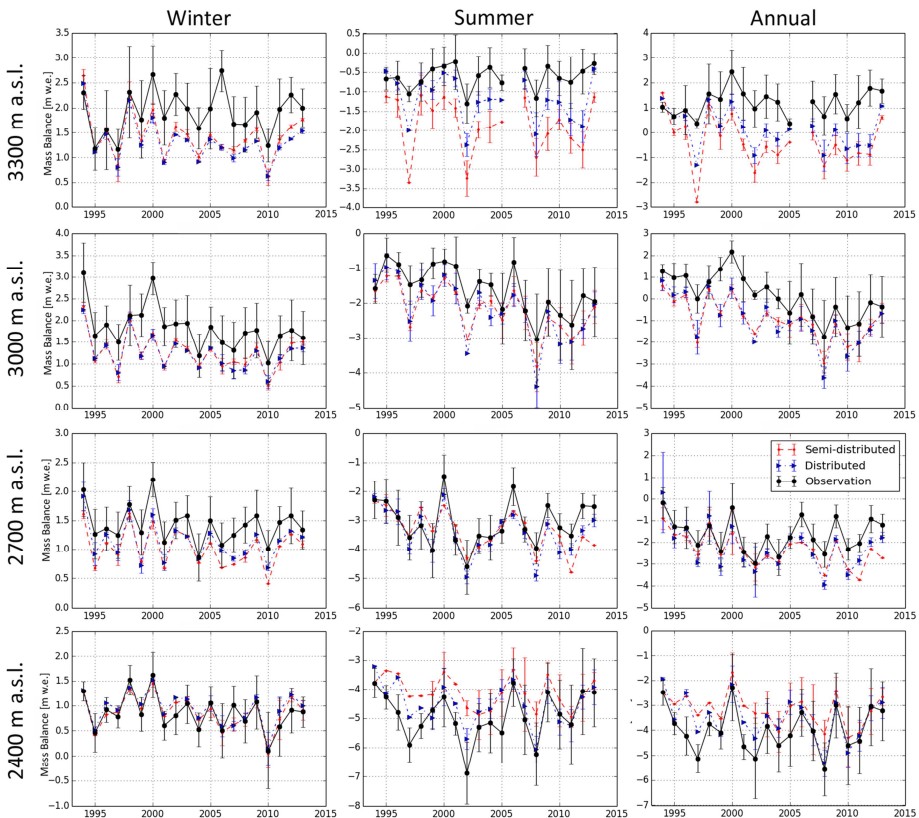


**Figure 10:** Temporal evolution of the observed and simulated (semi-distributed and
distributed) SMB for the Argentière glacier for the four 300-m elevation bands for the
period 1994–2013. The points show the average observation and simulation values for
the same measurement locations, and the vertical bars show the standard deviations for
those values.








**Figure 11:** Temporal evolution of the observed and simulated (semi-distributed and distributed) SMB for the Mer de Glace glacier for the seven 300-m elevations bands for the period 1994–2013. The points show the average observation and simulation values for the same measurement locations, and the vertical bars show the standard deviations for those values.





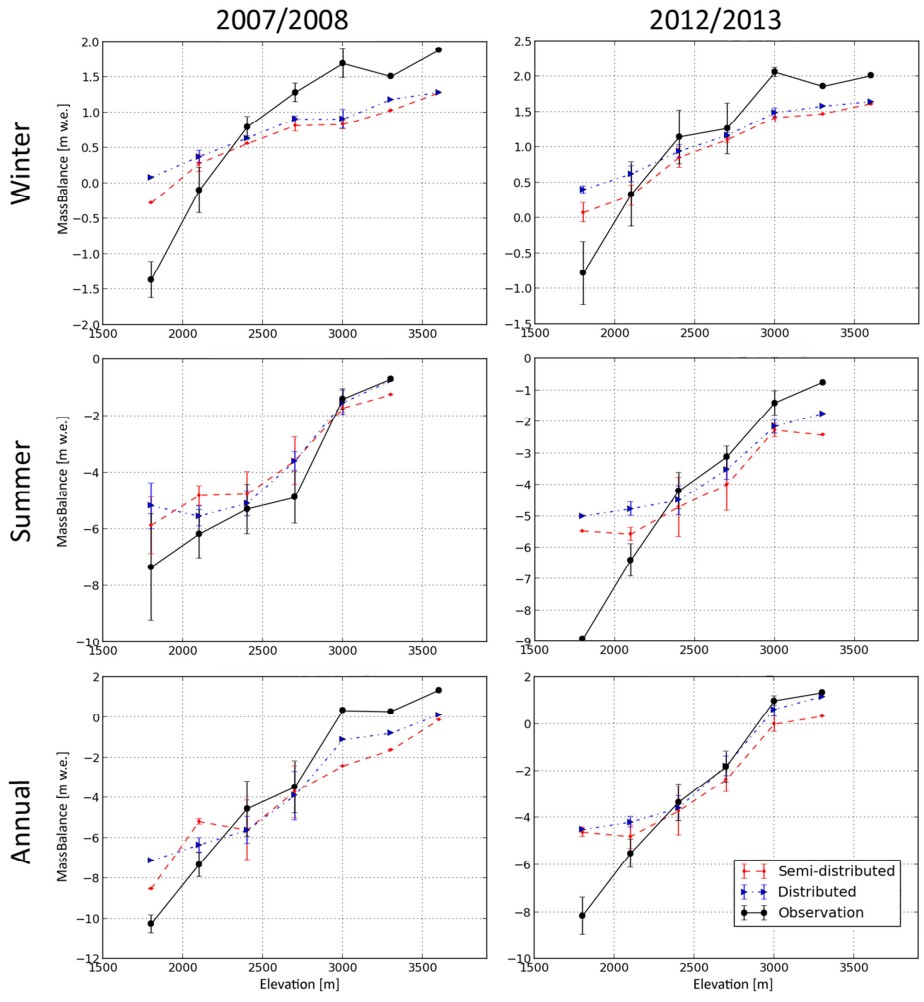

**Figure 12:** Altitudinal dependence of the observed and simulated (semi-distributed and distributed) SMB for two snow seasons (2007–08: low level snow accumulation; and 2012–13: high level snow accumulation) at the Mer de Glace glacier.



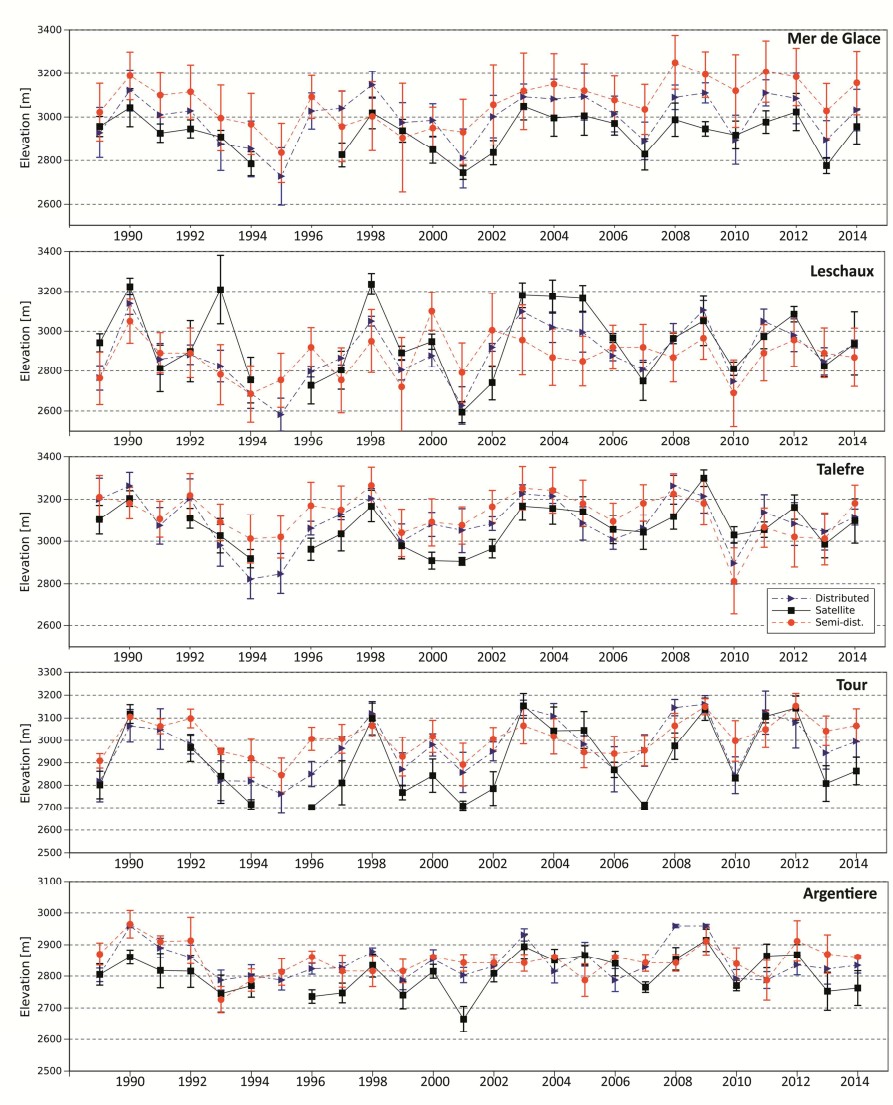

1233
**Figure 13:** Observed and simulated evolution of the ELA for the five glaciers during
the study period, based on the same dates as those for the satellite image acquisition.
1236





**Tables**

| Observatory | RMSE [cm] | Bias[cm] | Period | Num. Obs. |
|---|---|---|---|---|
| Chamonix | 23.3 | 12.1 | 1983-2015 | 6704 |
| Le Tour | 29.6 | 13.0 | 1985-2015 | 6323 |
| Nivose Aiguilles Rouges | 66.6 | 49.4 | 1983-2015 | 5902 |
| La Flegere | 45.0 | -19.1 | 2003-2015 | 1231 |
| Lognan | 20.8 | 1.9 | 1994-2015 | 5964 |

**Table 1**: Error statistics (bias and RMSE) between simulated and in situ snow depth observations for the five meteorological stations in the study area for periods for which observations were available. The locations of the stations are shown in Figure 1.

| Threshold SCA [0,1] | Threshold SD [m] | R2 | RMSE[cm] | MAE |
|---|---|---|---|---|
| | 0.1 | 0.821 | 12.64 | 8.36 |
| **0.35** | **0.15** | **0.828** | **12.51** | **8.24** |
| | 0.2 | 0.815 | 12.86 | 8.54 |

**Table 2**: UWS performance for various snow thicknesses selected as thresholds for the 2008–09 and 2009–10 snow seasons. Bold values indicate the selected snow depth threshold.

| Period | Approach | $R^2$ | MAE | RMSE |
|---|---|---|---|---|
| Entire period (2001–2015) | Semi-distributed | 0.815 | 10.47 | 15.28 |
| | Distributed | 0.822 | 8.35 | 12.64 |
| 2006–07 to 2007–08 | Semi-distributed | 0.744 | 10.756 | 16.903 |
| | Distributed | 0.756 | 8.74 | 14.82 |
| 2011–12 to 2012–13 | Semi-distributed | 0.881 | 11.56 | 15.58 |
| | Distributed | 0.895 | 7.99 | 11.10 |

**Table 3**: RMSE, MAE and $R^2$ values for the observed and simulated SCA (based on the distributed and semi-distributed approaches) for various time periods for the entire study area.



| Period | Approach | $R^2$ | MAE | RMSE |
|---|---|---|---|---|
| Entire period (2001–2015) | Semi-distributed | 0.71 | 10.12 | 16.04 |
| | Distributed | 0.72 | 7.60 | 12.84 |
| 2006–07 to 2007–08 | Semi-distributed | 0.58 | 11.26 | 18.36 |
| | Distributed | 0.59 | 8.61 | 15.62 |
| 2011–12 to 2012–13 | Semi-distributed | 0.82 | 11.30 | 16.38 |
| | Distributed | 0.84 | 7.79 | 11.69 |

**Table 4**: RMSE, MAE and $R^2$ values for the observed and simulated SCA (based on the
distributed and semi-distributed approaches) for various time periods for those parts of
the study area having a northern aspect (N, NE, NW).

| Period | Approach | $R^2$ | MAE | RMSE |
|---|---|---|---|---|
| Entire period (2001–2015) | Semi-distributed | 0.851 | 10.23 | 14.99 |
| | Distributed | 0.856 | 9.89 | 14.21 |
| 2006–07 to 2007–08 | Semi-distributed | 0.80 | 10.17 | 16.48 |
| | Distributed | 0.815 | 10.34 | 16.21 |
| 2011–12 to 2012–13 | Semi-distributed | 0.902 | 10.98 | 15.09 |
| | Distributed | 0.905 | 8.25 | 11.81 |

**Table 5**: RMSE, MAE and $R^2$ values for the observed and simulated SCA (based on the
distributed and semi-distributed approaches) for various time periods for those parts of
the study area having a southern aspect (S, SE, SW).




| Period | Approach | Jaccard | ASSD |
|---|---|---|---|
| Entire period (2001–2015) | Semi-distributed | 0.817 | 0.912 |
| | Distributed | 0.832 | 0.975 |
| 2006–07 to 2007–08 | Semi-distributed | 0.783 | 0.920 |
| | Distributed | 0.801 | 0.952 |
| 2011–12 to 2012–13 | Semi-distributed | 0.826 | 0.897 |
| | Distributed | 0.836 | 0.952 |

**Table 6:** Average values of the Jaccard index and ASSD values for each simulation
approach for various time periods.

| Period | Approach | Jaccard Index | | ASSD | |
|---|---|---|---|---|---|
| | | **JFM** | **MJJ** | **JFM** | **MJJ** |
| 2006–07 | Semi-distributed | 0.9535 | 0.802 | 0.687 | 1.152 |
| | Distributed | 0.9557 | 0.823 | 0.704 | 1.104 |
| 2007–08 | Semi-distributed | 0.950 | 0.793 | 0.717 | 1.062 |
| | Distributed | 0.951 | 0.809 | 0.724 | 1.043 |
| 2011–12 | Semi-distributed | 0.968 | 0.756 | 0.711 | 0.983 |
| | Distributed | 0.967 | 0.754 | 0.734 | 0.994 |
| 12012–13 | Semi-distributed | 0.980 | 0.790 | 0.199 | 1.271 |
| | Distributed | 0.990 | 0.799 | 0.198 | 1.250 |

**Table 7:** Average values of the Jaccard index and ASSD for each simulation approach
for the maximum (JFM) and minimum (MJJ) snow accumulation periods.



| Glacier | Period | Approach | RMSE | MAE | R2 | slope | Intersect |
|---|---|---|---|---|---|---|---|
| Arg | WSMB | Semi-distributed | 0.53 | 0.42 | 0.537 | 0.52 | 0.33 |
| | | Distributed | 0.52 | 0.40 | 0.51 | 0.458 | 0.467 |
| | SSMB | Semi-distributed | 0.96 | 0.78 | 0.72 | 0.56 | -1.47 |
| | | Distributed | 0.76 | 0.61 | 0.84 | 0.737 | -1.04 |
| | ASMB | Semi-distributed | 1.21 | 0.99 | 0.71 | 0.55 | -1.22 |
| | | Distributed | 1.05 | 0.85 | 0.78 | 0.679 | -1.02 |
| Mdg | WSMB | Semi-distributed | 0.72 | 0.56 | 0.64 | 0.53 | 0.093 |
| | | Distributed | 1.57 | 1.15 | 0.83 | 0.43 | 0.37 |
| | SSMB | Semi-distributed | 1.46 | 1.17 | 0.75 | 0.55 | -1.33 |
| | | Distributed | 1.19 | 0.86 | 0.86 | 0.67 | -0.94 |
| | ASMB | Semi-distributed | 1.72 | 1.33 | 0.75 | 0.52 | -1.45 |
| | | Distributed | 1.57 | 1.15 | 0.83 | 0.587 | -1.03 |

**Table 8:** RMSE, MAE, $R^2$ values for the slope and intersection in linear adjustments
between the observed and simulated SMB for Mer de Glace (Mdg) and Argentière
(Arg) glaciers.






| Glacier | Approach | Avg Dif | Std. Dev (Differences) | Slope | R2 |
|---------|----------|---------|------------------------|-------|-----|
| Mdg | Semi-distributed | 155.11 | 69.62 | 0.715 | 0.420 |
| | Distributed | 88.57 | 48.90 | 0.869 | 0.627 |
| Les | Semi-distributed | 158.34 | 101.84 | 0.188 | 0.102 |
| | Distributed | 110.73 | 109.67 | 0.560 | 0.586 |
| Tal | Semi-distributed | 105.14 | 59.25 | 0.4936 | 0.2336 |
| | Distributed | 80.12 | 41.87 | 0.766 | 0.476 |
| Tour | Semi-distributed | 105.14 | 59.25 | 0.339 | 0.528 |
| | Distributed | 84.33 | 68.71 | 0.625 | 0.715 |
| Arg | Semi-distributed | 63.89 | 42.87 | 0.270 | 0.103 |
| | Distributed | 54.52 | 31.85 | 0.578 | 0.381 |

**Table 9:** Average differences, standard deviations, slope of the linear adjustment, and
R2 values for the observed and simulated ELA for Mer de Glace (Mdg), Leschaux
(Les), Talefre (Tal), Tour and Argntière (Arg) glaciers.