# Peer review of "Distributed vs. semi-distributed simulations of snowpack 1 dynamics in alpine areas: case study in the upper Arve 2 catchment, French Alps, 1989–2015 3"

_The Cryosphere, 2017_

## Referee Comment (RC1) · Anonymous Referee #1 · 17 Oct 2017

**1 General comments**

**1.1 Summary of goals, approaches and conclusions**

Revuelto et al. studied the quality of a semi-distributed and a distributed energy balance snow cover model (Crocus) in an alpine catchment covering large differences in elevation and other topographic features. They were able to define an observation dataset which consists of satellite snow covered area (SCA), snow depth measured at five stations, glacier surface mass balance (SMB) data and the annual equilibrium line

altitude (ELA) of glaciers. Most of the validation data covers a period of 14 years. The semi-distributed approach integrates different elevation bands and aspects, while the distributed approach models shading, slope and elevation effects explicitly on a 250 m resolution. Meteorological forcing was obtained in the same 14 years' time period by a re-analysis model called SAFRAN, which delivers semi-distributed outputs. For the distributed version this data was interpolated to a grid.

Model output as snow depth, SCA, SMB and ELA was compared with observations using a variety of quality measures as root mean squared error, mean absolute error, and R-squared, amongst others. The authors also accounted for spatial similarities of the satellite derived spatial SCA products and modelled outputs.

The authors conclude that both approaches were able to reproduce observations, while there are slight advantages recognizable when using the distributed model approach.

1.2 Conceptual overview of the contribution of the paper

Evaluating different model approaches to model snow cover processes is important. Many research teams worldwide incorporate distributed or semi-distributed modelling approaches. Thus, it is helpful to evaluate a given modelling setup.

The aim of comparing two different model approaches based on the presented results is difficult to my opinion. My impression is that model deficiencies in both models may be able to override differences between them. Two model deficiencies could be mentioned. First, there is a scale issue between the meteorological forcing (massif scale 1000 km2) and the potential differences between the models (terrain shading effects). Second, both models handle SCA in a binary way using unrealistic thresholds for complex rough terrain.

Two other issues are the similarity to previously published papers, and the quite subjective validation procedure. Finally, the communication of the results is quite poor. Sometimes I am not able to follow statements in the text after studying tables and

figures, sometimes I am not able to follow the content of the sentences.

In the next subsections I will be more detailed to the here mentioned issues. I am convinced the authors are able to address my specific comments below. Thus I recommend to publish this manuscript in The Cryosphere after Major Revisions.

2 Specific comments

2.1 Potential model deficiencies

I understand that the authors want to establish a long data set for validation, which may have led to the decision to use SAFRAN with 14 years of meteorological forcing. However, after reading this manuscript I think SAFRAN does not vary precipitation amounts in one elevation/aspect category within the massif scale (i.e. 1000 km2). I am not sure if such a rough meteorological forcing is helpful to establish differences between a high resolution distributed model (250 m) and a semi-distributed model. I would suggest that a numerical weather model with a fine resolution (e.g. AROME with a 2.5 km resolution, Queno et al., 2016; or Vionnet et al., 2016) may be used as well for a few available years to show more clearly the potential of high-resolution modelling.

I am a bit surprised that an author who measured snow depth in alpine terrain in high resolution suggests a model approach that defines SCA to be zero, when modelled snow depth in a pixel is below 0.15 m, and one otherwise. 15 cm mean snow depth is not sufficient to cover large parts of a 250 m pixel in complex terrain. This may be true after a new snow fall, but not when a seasonal snow cover is melting out. While this is a common model approach, there were new SCA parameterisations developed in the past years to account for heterogeneous snow depth distribution based on terrain characteristics (e.g. Helbig et al., 2015; Cristea et al., 2017). This model evaluation has a large emphasis on SCA. In such a case I think a state-of-the-art SCA parameterisation is necessary. This is also necessary for a 250 m resolution grid since snow depth varies largely at smaller scales (e.g. Trujillo et al., 2007).

To my opinion, the benefit of terrain shading in the distributed approach may be overridden by these model deficiency.

**2.2 Validation**

The authors chose a quite subjective definition what a good model performance is. This lead to formulations as "appeared reliable" (line 439), "Overall, the ability [...] was satisfactory" (line 568ff). For many measures it is not clear to me if a difference in model performance is substantial or negligible. Here I would suggest a more objective procedure integrating a baseline model or quantifying error metrics in relation to year-to-year differences, for example.

**2.3 Differences and similarities to previous papers**

Last year this group of authors published two papers validating a SAFRAN/AROME-Crocus model (Queno et al., 2016; Vionnet et al., 2016). I would like to see more clearly what the advance of this manuscript is and how findings in those paper compare to this study (e.g. effects of meteorological forcing using SAFRAN). Some differences are obvious (250 m vs. 2.5 km resolution), long-term dataset with a spatial focus, but should be clearly mentioned in the Introduction.

**2.4 Communicating and presenting results**

I had large difficulties with understanding this manuscript. Statements in the text were sometimes not documented in the presented Figures and Tables. For example, I do not see in Figure 6 a higher consistency between observations and simulations in the winter in relation to the summer, which is stated in line 470. In Table 6 and 7 the ASSD values are higher for the distributed approach, but oppositely stated in line 506. Furthermore, I do not see that the distributed version is simulating the surface mass balance in the winter similarly well compared to the semi-distributed version (lines 757ff), while in Table 8 the RMSE for Mer-de-Glace and WSMB is much worse.

Many sentences remain unclear to me. Sometimes I am not able to determine if the
authors refer to winter or summer mass balance, or to which elevation band, or to what the demonstrative "this" is referring to. Improving the English grammar would certainly help to reach a clear and concise communication of the results.

3 Technical comments

The following comments show that some methods are not described in a sufficient way in order to be able to replicate results.

Lines 231ff: Please add more details on how the coarse SAFRAN categories were interpolated to the fine 250 m grid. The given citation (Vionnet et al., 2016) does not describe the procedure, but refers to another citation. Since energy balance models are quite sensitive to input data, the chosen interpolation methods should be presented here to understand the differences between the distributed and semi-distributed model approach.

Lines 291ff: Please add more details to the conclusion, why a full evaluation of the simulation is possible with the chosen validation data. What is a full evaluation and how do you apply Hanzer et al. (2016) on your dataset?

Line 325: I cannot find the name MODImLab in the two references. Is this a software built on these publications? Please clarify.

Line 329: I cannot find the term unmixing_wholesnow (UWS) in Charrois et al. (2013). Please clarify.

References

Cristea, N. C., Breckheimer, I., Raleigh, M. S., HilleRisLambers, J., & Lundquist, J. D. (2017). An evaluation of terrain‐based downscaling of fractional snow covered area datasets based on Lidar-derived snow data and orthoimagery. Water Resources Research, 53, 6802–6820, doi:10.1002/2017WR020799.

Helbig, N., van Herwijnen, A., Magnusson, J., & Jonas, T. (2015). Fractional snow-

covered area parameterization over complex topography. Hydrology and Earth System Sciences, 19(3), 1339-1351.

---

## Referee Comment (RC2) · Anonymous Referee #2 · 28 Oct 2017

The authors present a comprehensive evaluation and comparison of a snow model run in a semi-distributed and a fully distributed manner in an alpine basin in the French Alps. The model output is evaluated against an impressive collection of measurements including point-scale depth, glacier mass balance, glacier equilibrium elevation, and satellite-derived snow-cover area. The authors conclude that both models effectively simulate snow distribution over the study period, with the fully distributed version obtaining slightly better results.

The paper is well written and relatively easy to understand. The methodology and modeling is at the forefront of the field. The model application and verification is carefully crafted; however, many studies have successfully applied and validated a distributed model of seasonal snow and ice, so this aspect is not a scientific advancement. Rather, the novelty of the study is in the direct comparison of a semi-distributed and fully distributed snow model. This is an important and engaging science question.

I generally disagree with the interpretation of results and the conclusion that "... distributed simulations ... are the recommended modelling approach". Quite the opposite! I think the results make a case for the promotion of a semi-distributed snow model when carefully designed to parsimoniously maximize on relevant physiographical and meteorological information content while remaining computationally tractable.

In my opinion, the study comes up short of providing a comprehensive evaluation of the subjective modeling decisions and scaling issues that differentiate the two approaches. Thus, the paper misses an opportunity to offer a clear scientific advance. The authors present point-scale, semi-distributed, and fully distributed modeling as if they were three independent techniques with predefined structure. Rather, don't these approaches exist on a spectrum of scale and design, offering substantial flexibility to the modeler? Worse, there is little description of the authors' decision processes: 1) how were the # of semi-distributed units decided upon, 2) how was the 250 m grid scale of the distributed model determined (why not 350-m or 100-m), and 3) how sensitive might results be to these decisions?

The basic theory should be better explained in the Introduction with clear examples (mention unstructured grid design). I also missed a consideration of lateral flux exchange amongst grid-cells, which has been previously applied to both semi-distributed (i.e., HRU models) and fully distributed snow models. The authors state that data assimilation, snow transport, and shading of solar radiation treatment are not possible in a semi-distributed model configuration. This is incorrect and a more careful literature review must be conducted (e.g., MacDonald et al. (2009) for blowing snow; Marsh et al. (2012) for shading). This reasoning is the basis for the authors' conclusion that "...

distributed simulations . . . are the recommended modelling approach". The conclusion is unconvincing and unsupported by what little evidence is presented and discussed. In fact, the very topic stated in the title (Distributed vs. Semi-distributed) is not mentioned in the Discussion until the 5th page of that section.

The paper could be greatly improved. I encourage the authors to provide more theoretical background in the Introduction. In the Discussion, please thoroughly consider the subjective nature of model decisions (both generally and your own decisions) involved in the construction of a semi-distributed model. For example, could critical information (i.e., high-res. distributed forcing, satellite information, climatological information, and /or fully distributed model output) be leveraged to build a better semi-distributed model? The numerical parsimony offered by a semi-distributed model is not considered until Line 777!

A related issue that could be considered is the increasing need to assess potential climate change impacts on mountain cryosphere systems. This requires resolving snow and ice melt and river runoff at grid scales sufficient to resolve climate / elevation gradients, yet remaining computationally nimble to run extremely large ensembles for century-long historical and future periods. A semi-distributed model configuration could indeed help in this regard.

It is incomplete to evaluate a snow model against snow depth alone. A true and fair assessment should be conducted using snow water equivalent, which is a more relevant model state variable for water resources applications, and is a more direct evaluation of the energy balance. Are SWE measurements available in this region? Please include them.

References:

MacDonald, M. K., Pomeroy, J. W., & Pietroniro, A. (2009). Parameterizing redistribution and sublimation of blowing snow for hydrological models: tests in a mountainous subarctic catchment. Hydrological Processes, 23(18), 2570-2583.

Marsh, C. B., Pomeroy, J. W., & Spiteri, R. J. (2012). Implications of mountain shading on calculating energy for snowmelt using unstructured triangular meshes. Hydrological Processes, 26(12), 1767-1778.

Detailed Comments:

Abstract: Too much information on the methods . . . only two sentences on the results and conclusions.

Line 46: Mention atmospheric feedback?

Line 77: I'm not familiar with the term 'punctual' used in this manner. I prefer the term 'point-scale' as used in the Abstract.

Lines 102-105. This is incorrect. See my primary comments and references to HRU and TIN-based model application to blowing snow simulation and shading, respectively.

Line 106: The evaluation of performance shouldn't depend on the use . . . rather, determining which approach is optimal should depend on the use.

Lines 124-126: Shouldn't improved meteorological forcing fields also improve the meteorological forcing for semi-distributed / representative slopes?

Lines 132-135: This is not well explained. It may be better to state that the careful verification of the point-scale simulations helped to better interpret the results and comparisons between the semi-distributed and fully distributed models.

Line 208: Note in this section how many 1-D simulations were used in the semi-distributed run.

Lines 276-287: This entire section could be removed (not directly relevant to the results or conclusions).

Line 248: soil "moisture"? "humidity" suggests water in vapor phase.

Line 254: Explicitly state the specified ice thickness.

Lines 318-323: This information could be removed.

Line 388: the model doesn't technically "evolve" ice, but presumably only melts it.

Line 453: How is an underestimate inferred from this figure? I notice a 50% overestimate of snow depth in the Jan. event.

Line 476: "In winter the simulation . . .:" what simulation, exactly?

There are a number of 'in prep' citations, which are not relevant until published.

---

## Author Comment (AC1) · 22 Jan 2018

Author's comments:

Below we provide a detailed response to all comments and indicate resulting changes in the manuscript. Please, note that lines referred in this response are these of the manuscript tracked with changes. Additionally to changes referred in this response other changes have been accomplished to improve the final manuscript. For instance the manuscript title has been changed to better describe our study. Now the title is:

[Figure]

"Multi-criteria evaluation of snowpack simulations in complex alpine terrain with two spatialization approaches". Also some sections of the article have been reduced and others include further information; please check the manuscript with tracked changes to see them.

Reviewer's 1 comments: 2.1 Potential model deficiencies R1 I understand that the authors want to establish a long data set for validation, which may have led to the decision to use SAFRAN with 14 years of meteorological forcing. However, after reading this manuscript I think SAFRAN does not vary precipitation amounts in one elevation/aspect category within the massif scale (i.e. 1000 km2). I am not sure if such a rough meteorological forcing is helpful to establish differences between a high resolution distributed model (250 m) and a semi-distributed model. I would suggest that a numerical weather model with a fine resolution (e.g. AROME with a 2.5 km resolution, Queno et al., 2016; or Vionnet et al., 2016) may be used as well for a few available years to show more clearly the potential of high-resolution modelling.

Author's response (A): The SAFRAN meteorological reanalysis is used in this manuscript because it is still the best meteorological dataset available in the French Alps to drive spatialized snowpack simulations over a long period. The potential of the AROME Numerical Weather Prediction (NWP) model has already been investigated in other papers (Quéno et al., 2016; Vionnet et al., 2016). The spatial pattern of precipitation is more accurate with the kilometric resolution of AROME than with the massif scale of SAFRAN. However, some significant biases occur in AROME since the atmospheric forcing is only made of successive forecasts. No precipitation observations are assimilated to correct the precipitation amount, contrary to SAFRAN. As a result, both papers exhibit a better skill of snowpack simulations in terms of SWE and snow depth when SAFRAN is used as forcing. In the case of the Mont-Blanc massif, unpublished results (Vionnet et al, in prep.) were obtained for the "Mer de Glace" glacier for winter 2011/2012 and are provided here. Figure 1 shows the glacier winter surface mass balance (SMB) simulated with Crocus and driven by three meteorological

forcing: AROME at 2.5 km downscaled to 0.5 km (green), an experimental version of AROME at 0.5 km resolution (blue) and SAFRAN spatially distributed at 0.5 km resolution (red). This figure also includes winter SMB observations (black dots) and their associated uncertainty. At high elevations where there is no observation assimilated by SAFRAN, the winter SMB is underestimated with SAFRAN and is more realistic with AROME. However, at lower elevations ($\sim$ below 2500 m) the winter SMB is highly overestimated with AROME and more realistic with SAFRAN thanks to the assimilation of observations. Therefore, we decided to use the atmospheric forcing from SAFRAN in our study since AROME still suffers from limitations in the Mont Blanc area. One of the goals of this paper is also to study the impact of shadowing in high spatial resolution snowpack simulations regardless the future possible availability of a higher resolution meteorological forcing. Therefore, it makes sense to perform this analysis with the same meteorological dataset in both simulations and to choose the best and longest dataset currently available for that purpose.

Figure 1 caption (complete): Glacier Winter Surface Mass Balance observed and simulated for the "Mer de Glace" in 2011-2012 snow season. SAF correspond to Crocus simulations with SAFRAN meteorological forcing, ARO_0p5 to and ARO_2p5D to Crocus simulations driven respectively by an experimental version of AROME at 0.5 km and AROME 2.5 km downscaled at 0.5 km and Observations correspond to punctual observations. For each mass balance a linear tendency line has been included.

Obviously, taking benefit from high resolution NWP models in snowpack simulations is still a major research perspective. This still requires the development of a high-resolution distributed analysis combining observations and AROME forecast and the development of downscaling methods to fill the gap between their kilometric resolution and the 250 m resolution which is necessary for an accurate representation of slopes in alpine environments. Please refer to response to the comment 2.3 for the corresponding changes in the manuscript.

R1: I am a bit surprised that an author who measured snow depth in alpine terrain in

high resolution suggests a model approach that defines SCA to be zero, when modelled snow depth in a pixel is below 0.15 m, and one otherwise. 15 cm mean snow depth is not sufficient to cover large parts of a 250 m pixel in complex terrain. This may be true after a new snow fall, but not when a seasonal snow cover is melting out. While this is a common model approach, there were new SCA parameterisations developed in the past years to account for heterogeneous snow depth distribution based on terrain characteristics (e.g. Helbig et al., 2015; Cristea et al., 2017). This model evaluation has a large emphasis on SCA. In such a case I think a state-of-the-art SCA parameterisation is necessary. This is also necessary for a 250 m resolution grid since snowdepth varies largely at smaller scales (e.g. Trujillo et al., 2007). To my opinion, the benefit of terrain shading in the distributed approach may be overridden by these model deficiency.

A: We agree that considering 0.15 m as the threshold for considering a simulated pixel as snow covered or not, may be disputable if inferring full snow cover in the pixel, especially considering the high spatial heterogeneity of snowpack in mountain areas. This is also true for the choice of threshold used to convert SCA derived from MODIS observations to a binary map for use of the selected metrics. However, most of the works which studied the snow covered area over large domains use a threshold for considering a pixel as snow covered or not , regardless the spatial scale(e.g. Cristea et al., 2017 at the very high resolution of 3 m, Gascoin et al., 2015 at the resolution of 500 m). Furthermore, we demonstrate in our paper that the strict validity of this relationship is secondary in the context of the evaluations performed in this paper. Indeed, we calculated the evaluation metrics for several SD and SCA threshold and the results are now included in Table 2 in the revised manuscript. The calculations were obtained for both spatialization approaches. This table shows that the evaluation metrics are only slightly sensitive to the choice of these thresholds and that for every threshold and metrics the ranking of the two approaches remains the same. The selected thresholds are those leading to the best scores. Note also that the snow depth threshold of 15 cm is consistent with other studies (e.g., Figure 4 in Gascoin et al. 2015). Testing fractional SCA parametrizations hypotheses would require higher spatial resolution simulation, which could equally be argued to involve a higher uncertainty in relation to the approach involved for spatial distributing (e.g. precipitation quantities must be changed depending on the topographic characteristics). Moreover fractional SCA area hypotheses would also require a higher spatial resolution database to test the results regarding observations of snow presence/absence (Cistera et al., 2017) or snow depth (Helbig et al., 2015), commonly derived from LiDAR observations. Additionally, testing higher spatial resolution snowpack simulations also requires significantly more computational time. An increase on the spatial resolution from 250 m to about 20-30 m pixel size would be the only alternative that may potentially reproduce a realistic snow distribution in mountain terrain (Deems et al., 2006, Trujillo et al., 2007, Revuelto et al., 2014) which is beyond the scope of the present study. The following changes were consequently performed in the manuscript (please also check manuscript with tracked changes for smaller changes). Lines 565-573, "Table 2 shows the SCA simulation results estimated based on 0.1, 0.15 and 0.2 m snow depth thresholds compared with the various UWS thresholds tested, for the 2008–09 and 2009–10 snow seasons (average snow accumulations) and for both spatialization approaches. This table shows that the evaluation metrics are only slightly sensitive to the choice of these thresholds and that for every threshold and metrics the ranking of the two approaches remains the same. In light of the sensibility test results we selected a 0.15 m snow depth threshold for the simulations and 0.35 SCA threshold for MODImLab UWS product for classifying a pixel as snow-covered. Lines 744-757: "Crocus simulates the energy and mass exchanges with soil and atmosphere and also within the snowpack layers, but it does not simulate small scale topographic effects on snow depth distribution (Revuelto et al., 2016a). Given the fact that the final objective of this study is to compare two simulation approaches and that one of them would not allow an appropriate parametrization of topographic control on snow distribution (semi-distributed approach), we have not considered novel approaches for distributing snow based on terrain parameters (Cristera et al., 2017, Helbig et al., 2015), which may also require a higher spatial resolution

for accounting topographic effect on snow distribution (Deems et al., 2006, Trujillo et al., 2007). Hence, we decided to simulate snowpack evolution with a spatial scale in which satellite observations were available over a long time period with a suitable temporal resolution, which lead to select 250m spatial resolution simulations (same as MODImLab products). Moreover this spatial resolution provides an appropriate representation of slopes for future applications forecasting snow avalanches with expert systems (MEPRA, Lafaysse et al., 2013)."

R1: 2.2 Validation The authors chose a quite subjective definition what a good model performance is. This lead to formulations as "appeared reliable" (line 439), "Overall, the ability [: : :] was satisfactory" (line 568ff). For many measures it is not clear to me if a difference in model performance is substantial or negligible. Here I would suggest a more objective procedure integrating a baseline model or quantifying error metrics in relation to year to year differences, for example.

A: We agree with the reviewer that the comparison of scores was not sufficiently based on objective criteria in the initial manuscript. In order to provide a more objective quantification of error metrics (RMSE, MAE and R2) a 100-member bootstrap obtained by random sampling with replacement of the different years of observations (14 for the SCA evaluation and 20 for the SMB evaluation) was generated. This bootstrap allows us assessing the uncertainty of each score. Thus, we can compare the scores of the semi-distributed and distributed simulations by a classical t-student test: the associated p-value obtained from the bootstrapped sample allows us to accept or reject the null hypothesis (i.e. equality of scores). Tables 3 and 8, which previously showed only error metrics averages for the whole period, now show the average value and the standard deviation of RMSE, MAE and R2 obtained from the bootstrapped sample. Error metrics in bold note p-values lower than 0.01 (99% confidence interval for rejecting null hypothesis). The standard deviations of the RMSE and the MAE for the snow cover area are lower than the difference between the scores of both approaches. As a result, the p-value allows us to reject the null hypothesis with a 99% of confidence interval and

that the skills of distributed and semi-distributed simulations are not statistically equivalent. Conversely, the standard deviation of the R2 for the snow cover area is high compared to the difference between the scores of the two spatial discretizations. As a result, the high p-value indicated that the null hypothesis should be accepted and that these scores are not statistically different between both approaches. More generally, when looking at all scores for SCA and SMB, we can conclude objectively from this new analysis that there is a slight but significant added value of the distributed discretization. Finally, formulations as these noted by the reviewer have been removed in final manuscript version. Please check the manuscript with tracked changed to see how are described the bootstrapping and the t-student test. Result section now includes the following lines describing the results obtained: Lines 602-619: "Error estimates for the SCA simulated for the whole study site and for north and south aspects (Tables 3, 4 and 5) were lower for the distributed simulations compared with the satellite observations. RMSE and MAE standard deviations obtained from the bootstrapping (Table 3) are lower than the difference between scores for both approaches. The p-values for these two error metrics are lower than 0.01 and thus the null hypothesis is rejected with a 99% confidence interval and the skills of distributed and semi-distributed simulations are not statistically equivalent. Conversely the R2 standard deviation of the SCA is high compared to the difference between the scores of both approaches. As a result, the high p-value indicated (in this case above 0.05) that the null hypothesis should be accepted and that these scores are not statistically different between both approaches. R2, MAE and RMSE average values for high (2006-2008 snow seasons, Table 4) and low (2011-2013 snow seasons, Table 5) levels of snow accumulation also show the better capacity of distributed simulations to reproduce SCA evolution. The t-student test has demonstrated that RMSE and MAE results for both approaches are not statistically equivalent and that for all aspects and periods, the distributed simulations presents lower errors. We can conclude that this latter approach significantly better reproduce the SCA evolution. The differences in the error metrics (RMSE and MAE) between distributed and semi-distributed simulations are significant for both, north and

south aspects but higher for north aspect. However, it must be highlighted that for the whole catchment and for any aspect, the null hypothesis can be accepted based on the R2 value between distributed and semi-distributed approaches. This means that the added value of the distributed approach is not visible on this criterion." Lines 703-710: "Table 8 shows RMSE, MAE and R2 means and standard deviations obtained from the 100-member bootstrap sample. For most of the error metrics, the standard deviations are lower than score differences and the p-values are low enough to reject null hypothesis. This way results obtained with both simulation approaches are statistically different. In winter, the SMB simulations show similar results. For both glaciers, lower RMSE and MAE are obtained for distributed simulations and better R2 for semi-distributed simulations. Oppositely during summer all scores show better results for the distributed approach. The annual SMB also exhibit better results for distributed simulations."

R1: 2.3 Differences and similarities to previous papers Last year this group of authors published two papers validating a SAFRAN/AROMECrocus model (Queno et al., 2016; Vionnet et al., 2016). I would like to see more clearly what the advance of this manuscript is and how findings in those papers compare to this study (e.g. effects of meteorological forcing using SAFRAN). Some differences are obvious (250 m vs. 2.5 km resolution), long-term dataset with a spatial focus, but should be clearly mentioned in the Introduction.

A: Following the reviewer recommendation, we included a new paragraph in the introduction. This paragraph describes the main characteristics of the simulations obtained in Vionnet et al., 2016 and Queno et al., 2016. Additionally, the main objectives of this work are contextualized within the findings of these two works. The next lines have been included in the introduction (lines 147-180): "Recent studies have assessed the impact of high-resolution atmospheric forcing from the Numerical Weather Prediction system AROME (Seity et al., 2010) on distributed snowpack simulations with Crocus. Queno et al., (2016) and Vionnet et al., (2016) compared simulations at a 2.5 km spatial resolution forced by AROME forecasts or by SAFRAN reanalysis (Durand et al., 2009). These works demonstrated that the geographical patterns simulated by the AROME-Crocus model chain are realistic and more detailed than the SAFRAN-Crocus model chain over large areas (the Pyrenees and French Alps). Nevertheless these studies also exhibit some significant biases in meteorological and snow variables with the AROME-Crocus chain which do not assimilate any meteorological observation, in particular precipitation. As a result, Queno et al., (2016) and Vionnet et al., (2016) exhibit a better skill of snowpack simulations when SAFRAN is used as forcing. They conclude that the potential of the high spatial resolution atmospheric forcing from the NWP system will be more beneficial in snowpack simulations with the development of a high-resolution distributed analysis combining observations and AROME forecast and the development of downscaling methods to fill the gap between their kilometric resolution and the resolution required to capture the variability of slopes and aspects in alpine environments. Moreover the impact of topographic effects on snowpack simulations (implemented in the snowpack model) has not yet been assessed in detail. At present, the implementation of terrain shadowing effects on Crocus snowpack model (achieved on distributed simulations) has not been analyzed in complex alpine terrain. It is therefore necessary to compare distributed and semi-distributed snowpack simulations with a spatial resolution that enables a detailed representation of alpine terrain."

R1: 2.4 Communicating and presenting results I had large difficulties with understanding this manuscript. Statements in the text were sometimes not documented in the presented Figures and Tables. For example, I do not see in Figure 6 a higher consistency between observations and simulations in the winter in relation to the summer, which is stated in line 470. In Table 6 and 7 the ASSD values are higher for the distributed approach, but oppositely stated in line 506. Furthermore, I do not see that the distributed version is simulating the Surface mass balance in the winter similarly well compared to the semi-distributed version (lines 757ff), while in Table 8 the RMSE for Mer-de-Glace and WSMB is much worse.

A: The way of communicating results was clarified in the revised paper. Moreover the new statistical framework included comparing the scores of the distributed and semi-distributed simulations of SCA and SMB allows obtaining more robust conclusion. The last remark about previous version of Table 8 regarding the WSMB was due to an error on the values previously shown. Now this table presents the bootstrapping averages and standard deviation of error metrics for both approaches, for Mdg the difference between the semi-distributed and the distributed approaches for WSMB is smaller than for Arg Moreover the MAE of both approaches for the WSMB in Mer de Glace is not significantly different. Thus the statement has been deleted in the revised manuscript.

R1: 3 Technical comments The following comments show that some methods are not described in a sufficient way in order to be able to replicate results. Lines 231ff: Please add more details on how the coarse SAFRAN categories were interpolated to the fine 250 m grid. The given citation (Vionnet et al., 2016) does not describe the procedure, but refers to another citation. Since energy balance models are quite sensitive to input data, the chosen interpolation methods should be presented here to understand the differences between the distributed and semi-distributed model approach.

A: We included the following complementary information to describe the interpolation technique, (lines 309-318): "As SAFRAN reanalysis provides semi-distributed outputs, the meteorological forcing at hourly time steps was spatially distributed over the 250-m grid DEM using specific routines that accounted for the elevation and aspect of each grid cell. For each cell of the 250-m grid, the spatialization of meteorological variables from the 300-m elevation bands of SAFRAN is based on a linear interpolation between the two closest elevation bands. Only one SAFRAN aspect class is considered for each pixel (nearest-neighbour technique for the aspect) (Vionnet et al., 2016). Therefore, the meteorological input data are similar for all simulations: only minor differences occur because elevation differences (< 300 m) may impact meteorological forcing variables.

R1: Lines 291ff: Please add more details to the conclusion, why a full evaluation of the simulation is possible with the chosen validation data. What is a full evaluation and

how do you apply Hanzer et al. (2016) on your dataset?

A: We base our statement about the full evaluation of simulations on the representation Hanzer et al., (2016) introduced and the database they used on their evaluation. From the eight datasets they considered we used four of them, which combined showed a validation dataset quite close to their evaluation. This issue has been clarified in the text as follows (lines 368-377): "To evaluate the simulations in this study we used four datasets based on: in situ snow depth from Météo-France stations; the snow covered area (SCA) from MODIS images; the punctual glacier surface mass balance (SMB); and the glacier equilibrium-line altitude (ELA) from Landsat/SPOT/ASTER. Based on the radar charts presented by Hanzer et al. (2016), shown in their Figure 5, the information available for our study matches four of the datasets exploited in their study (Snow Depth, MODIS, Landsat and Glacier mass balance). These four datasets cover almost the full radar chart space ("optimal" validation dataset), thus providing almost a full evaluation of the simulation performance."

R1: Line 325: I cannot find the name MODImLab in the two references. Is this a software built on these publications? Please clarify.

A: MODImLab software is based on the research of these two publications. This is now stated in the text. The reference to MODImLab user manual has also been added in the manuscript.

R1: Line 329: I cannot find the term unmixing_wholesnow (UWS) in Charrois et al. (2013). Please clarify.

A: We have clarified that the UWS product we present from MODImLab user's manual corresponds to the linear unmixing technique introduced in Charrois et al., (2013)

References:

Charrois, L., Dumont, M., Sirguey, P., Morin, S., Lafaysse, M., and Karbou, F. (2013). Comparing different MODIS snow products with distributed distributed simulation of the

snowpack in the French Alps. Proceedings of the International Snow Science Workshop Grenoble – Chamonix Mont-Blanc - 2013 (Grenoble, France), 937-941

Cristea, N. C., Breckheimer, I., Raleigh, M. S., HilleRisLambers, J., and Lundquist, J. D. (2017). An evaluation of terrainâËŸARËǦ based downscaling of fractional snow covered area datasets based on Lidar-derived snow data and orthoimagery. Water Resources Research, 53, 6802–6820,

Durand, Y., Giraud, G., Brun, E., Merindol, L., and Martin, E. (1999). A computer-based system simulating snowpack structures as a tool for regional avalanche forecasting. J. Glaciol. 45, 469–484.

Durand, Y., Laternser, M., Giraud, G., Etchevers, P., Lesaffre, B., and Mérindol, L. (2009). Reanalysis of 44 Yr of Climate in the French Alps (1958–2002): Methodology, Model Validation, Climatology, and Trends for Air Temperature and Precipitation. J. Appl. Meteorol. Climatol. 48, 429–449.

Deems, J. S., Fassnacht, S. R., & Elder, K. J. (2006). Fractal distribution of snow depth from LiDAR data. Journal of Hydrometeorology, 7(2), 285-297.

Gascoin, S., Hagolle, O., Huc, M., Jarlan, L., Dejoux, J.-F., Szczypta, C., Marti, R., and Sánchez, R. (2015). A snow cover climatology for the Pyrenees from MODIS snow products. Hydrol Earth Syst Sci 19, 2337–2351.

Hanzer, F., Helfricht, K., Marke, T., and Strasser, U. (2016). Multilevel spatiotemporal validation of snow/ice mass balance and runoff modeling in glacierized catchments. The Cryosphere 10, 1859–1881.

Helbig, N., van Herwijnen, A., Magnusson, J., and Jonas, T. (2015). Fractional snow-covered áreas parametrization over complex topography. Hydrology and Earth System Sciences, 19(3), 1339-1351.

Lafaysse, M., Morin, S., Coléou, C., Vernay, M., Serça, D., Besson, F., Willemet, J.M., Giraud, G., and Durand, Y. (2013). Towards a new chain of models for avalanche

hazard forecasting in French mountain ranges, including low altitude mountains. Int. Snow Sci. Workshop Grenoble-Chamonix Mont-Blanc.

Quéno, L., Vionnet, V., Dombrowski-Etchevers, I., Lafaysse, M., Dumont, M., and Karbou, F. (2016). Snowpack modelling in the Pyrenees driven by kilometric-resolution meteorological forecasts. The Cryosphere 10, 1571–1589.

Revuelto, J., López-Moreno, J.I., Azorin-Molina, C., and Vicente-Serrano, S.M. (2014). Topographic control of snowpack distribution in a small catchment in the central Spanish Pyrenees: Intra- and inter-annual persistence. Cryosphere 8, 1989–2006.

Seity, Y., Brousseau, P., Malardel, S., Hello, G., Bénard, P., Bouttier, F., Lac, C., and Masson, V. (2010). The AROME-France Convective-Scale Operational Model. Mon. Weather Rev. 139, 976–991.

Trujillo, E., Ramírez, J. A., & Elder, K. J. (2007). Topographic, meteorologic, and canopy controls on the scaling characteristics of the spatial distribution of snow depth fields. Water Resources Research, 43(7).

Vionnet, V., Dombrowski-Etchevers, I., Lafaysse, M., Quéno, L., Seity, Y., and Bazile, E. (2016). Numerical Weather Forecasts at Kilometer Scale in the French Alps: Evaluation and Application for Snowpack Modeling. J. Hydrometeorol. 17, 2591–2614.

Vionnet, V., Six, D., Auger, A., Lafaysse, M., Queno, L., Reveillet, M., Dombrowski-Etchevers, I., Thibert, E. and Dumont, M. (in prep). Influence of precipitation datasets on sub-kilometric simulations of snowpack and glacier winter mass balance in alpine terrain.
* * *
**Mer de Glace**

Mass balance (mm w.e.)

Elevation (m)

Legend:
- SAF
- ARO_0p5
- ARO_2p5D
- Observations

**Fig. 1.** Glacier Winter Surface Mass Balance observed and simulated for the "Mer de Glace" in 2011-2012 snow season. SAF correspond to Crocus simulations with SAFRAN meteorological forcing, ARO_0p5 to and ARO_

---

## Author Comment (AC2) · 22 Jan 2018

Author's comments:

Below we provide a detailed response to all comments, and indicate resulting changes in the manuscript. Please, note that lines referred in this response are these of the manuscript tracked with changes. Additionally to changes referred in this response other changes have been accomplished to improve the final manuscript. For instance the manuscript title has been changed to better describe our study. Now the title is:

"Multi-criteria evaluation of snowpack simulations in complex alpine terrain with two spatialization approaches". Also some sections of the article have been reduced and others include further information; please check the manuscript with tracked changes to see them.

Reviewer 2 (R2) I generally disagree with the interpretation of results and the conclusion that ". . . distributed simulations. . . are the recommended modelling approach". Quite the opposite! I think the results make a case for the promotion of a semi-distributed snow model when carefully designed to parsimoniously maximize on relevant physiographical and meteorological information content while remaining computationally tractable.

A: The comparison between both approaches has been improved in the revised manuscript (see response to comment 2.2 from Reviewer 1). In particular, a bootstrap approach is now used to test the significance of the differences between the scores. Thus, we can now provide an objective comparison of simulation results. Furthermore, despite the statistically significant improvement obtained with the distributed simulations, we agree with Reviewer 2 that the skill of the semi-distributed simulations is sufficient in many applications with much lower computational requirements. Therefore, following the reviewer's comment, the abstract and the conclusions of the paper were modified as follows: Abstract final sentence: "Slightly better results were obtained using the distributed approach. The improvement is statically significant mainly because it includes the effects of shadows and terrain characteristics (local values of aspect, slope and elevation for each grid cell). However, the minor improvement observed with a much higher computational time does not justify the recommendation of this approach for all applications as long as distributed simulations are not combined with new data assimilation techniques and higher-resolution meteorological inputs. " Similarly in the conclusions: "Overall, the results of this study demonstrated that distributed simulations reproduce slightly better snowpack dynamics in the alpine terrain of our study area. Distributed simulations take into account the specific topographic characteristics

of each pixel (local values of aspect, slope and elevation) and more importantly the effects of terrain shadowing by surrounding areas. Accounting for these two effects over long time periods led to statistically significant better results for the distributed approach. However the lower computational requirements of semi-distributed simulations together with the flexibility on the design and application scale of the simulation make this approach also suitable to simulate snowpack evolution."

R2: In my opinion, the study comes up short of providing a comprehensive evaluation of the subjective modeling decisions and scaling issues that differentiate the two approaches. Thus, the paper misses an opportunity to offer a clear scientific advance. The authors present point-scale, semi-distributed, and fully distributed modeling as if they were three independent techniques with predefined structure. Rather, don't these approaches exist on a spectrum of scale and design, offering substantial flexibility to the modeler? Worse, there is little description of the authors' decision processes: 1) how were the # of semi-distributed units decided upon, 2) how was the 250 m grid scale of the distributed model determined (why not 350-m or 100-m), and 3) how sensitive might results be to these decisions?

A: 1) The design of the semi-distributed approach corresponds in terms of elevation, aspect and slope classes correspond to the design of the operational system used for avalanche hazard forecasting in France for more than 20 years (Durand et al, 1999 ; Lafaysse et al 2013), as mentioned in the paper in section 3.1 2) and 3) We have better described in the introduction why the 250 m spatial resolution was chosen and we have also discussed which consequences these decision may have on results. Here are the new sentences included in the introduction (Line 182-189): "...The final products of both simulations are 250 m gridded snowpack datasets. This spatial resolution was selected because it renders slopes sufficiently well to describe small valleys with significant shadowing effects. It will also allow to explore snow mechanical stability in future avalanche hazard forecasting applications. Indeed broader resolutions imply a too strong smoothing of terrain to represent slopes steep enough for avalanche release. The 250 m grid cell size of the simulations also enables a direct comparison with optical satellite products at the same spatial resolution. ….." Moreover, the discussion also includes the following lines addressing this issue (lines 925-931) "The results obtained in this study, i.e. slightly but significantly better skill for the distributed approach, are sensitive to the choice of the spatial resolution. Using resolution coarser than 250 m would lead to smaller differences between both spatialization approaches because the pixel elevations would be less accurate and because all the shadows would not be resolved. Conversely, higher resolutions may improve the accuracy of shadowing effects but with computational times which can become unaffordable for large areas applications."

R2: The basic theory should be better explained in the Introduction with clear examples (mention unstructured grid design). I also missed a consideration of lateral flux exchange amongst grid-cells, which has been previously applied to both semi-distributed (i.e., HRU models) and fully distributed snow models. The authors state that data assimilation, snow transport, and shading of solar radiation treatment are not possible in a semi-distributed model configuration. This is incorrect and a more careful literature review must be conducted (e.g., MacDonald et al. (2009) for blowing snow; Marsh et al. (2012) for shading). This reasoning is the basis for the authors' conclusion that "…distributed simulations… are the recommended modelling approach". is unconvincing and unsupported by what little evidence is presented and discussed. In fact, the very topic stated in the title (Distributed vs. Semi-distributed) is not mentioned in the Discussion until the 5th page of that section.

A: The reference to an "unstructured grid design" is now mentioned in line 85 and 297. We have also conducted a more comprehensive literature review including the articles cited by Reviewer 2 and some more. In this regard the introduction now includes more theoretical background and the discussion was also improved on how snow transport and terrain shadowing could be included on semi-distributed simulations. Moreover the order of the different sections of the discussion was changed. The possibility to

implement satellite data assimilation and blowing snow schemes in semi-distributed approaches are now detailed in the introduction (lines 129-140): "Semi-distributed simulations may also allow the implementation of satellite data assimilation techniques (Mary et al, 2013) but they would require specific routines for aggregating observations and would reduce potential benefits of high resolution satellite observations. Similarly, blowing snow can be simulated in the semi-distributed approach (MacDonald et al., 2009, Vionnet et al., 2018). Vionnet et al., (2018) show strong assumptions on the topography are necessary to transport snow mass from one aspect to another (virtual ridge between opposite aspect classes for any elevation band). In MacDonald et al., (2009), the model parametrization requires a discretization of the study site based on a strong knowledge of the area from previous works (McCartney et al., 2006, Pomeroy et al., 1999, 2006). Thus the transferability of these results to large domains for which detailed information on the landscape features is not available is questionable." The representation of shadows in intermediate spatial discretization is now discussed (line 1047-1052): "Other approaches halfway between our distributed and semi-distributed snowpack simulations are also showing promising results. This is the case of unstructured triangular meshes, which allow better capturing horizon- shadows of surrounding topography than the semi-distributed approach used in this work. These methods are able to improving energy balance simulation results while preserving computational costs (Marsh et al., 2012)." Finally we included a short sentence regarding lateral flux exchanges not implemented in Crocus snowpack model (lines 900-902): "Similarly other processes such as lateral heat flux exchanges amongst grid-cells are not implemented in Crocus snowpack model and thus could impact the final result of simulations (Harder and Pomeroy 2017)."

R2: The paper could be greatly improved. I encourage the authors to provide more theoretical background in the Introduction. In the Discussion, please thoroughly consider the subjective nature of model decisions (both generally and your own decisions) involved in the construction of a semi-distributed model. For example, could critical information (i.e., high-res. distributed forcing, satellite information, climatological information, and /or fully distributed model output) be leveraged to build a better semi-distributed model? The numerical parsimony offered by a semi-distributed model is not considered until Line 777! A: In the response to the previous comment, we illustrate how we improved the introduction and the discussion to discuss the possibilities to improve a semi-distributed approach by this different potential complementary information. The discussion section now incorporates a boarder analysis on the impact of model decisions and the possibility of developing better semi-distributed models integrating; satellite data, distributed forcing etc.: The numerical parsimony is now mentioned earlier in the discussion (line 877 in the document with tracked changes) as a main advantage of the semi-distributed approach and better emphasized in the abstract and the conclusion.

R2: A related issue that could be considered is the increasing need to assess potential climate change impacts on mountain cryosphere systems. This requires resolving snow and ice melt and river runoff at grid scales sufficient to resolve climate / elevation gradients, yet remaining computationally nimble to run extremely large ensembles for century-long historical and future periods. A semi-distributed model configuration could indeed help in this regard.

A: Discussion section, now also presents a short discussion on the importance of using semi-distributed simulations to analyse the impact of climate change on mountain areas. Section 5.2 now ends with the following paragraph (line 883-885): "A good example of an application in which the computational requirements have a determinant weight are ensemble simulations for projections in several climate scenarios (e.g. Verfaille et al, 2017). "

R2: It is incomplete to evaluate a snow model against snow depth alone. A true and fair assessment should be conducted using snow water equivalent, which is a more relevant model state variable for water resources applications, and is a more direct evaluation of the energy balance. Are SWE measurements available in this region? Please include them.

A: We agree with Reviewer 2 that it is incomplete to evaluate a snow model with only snow depth observations. However, a large number of Snow Cover Area and Snow Water Equivalent (i.e. glacier Surface Mass Balance) measurements are included in the evaluation datasets (see description in section 3.4). Note that unfortunately for the five stations used in the punctual evaluation, only SD measurements are available.

References:

Charrois, L., Dumont, M., Sirguey, P., Morin, S., Lafaysse, M., and Karbou, F. (2013). Comparing different MODIS snow products with distributed distributed simulation of the snowpack in the French Alps. Proceedings of the International Snow Science Workshop Grenoble – Chamonix Mont-Blanc - 2013 (Grenoble, France), 937-941

Cristea, N. C., Breckheimer, I., Raleigh, M. S., HilleRisLambers, J., and Lundquist, J. D. (2017). An evaluation of terrainâĚŸARËĞ based downscaling of fractional snow covered area datasets based on Lidar-derived snow data and orthoimagery. Water Resources Research, 53, 6802–6820,

Durand, Y., Giraud, G., Brun, E., Merindol, L., and Martin, E. (1999). A computer-based system simulating snowpack structures as a tool for regional avalanche forecasting. J. Glaciol. 45, 469–484.

Harder, P., Pomeroy, J. W. and Helgason, W. (2017) "Local Scale Advection of Sensible and Latent Heat During Snowmelt." Geophysical Research Letters 44.19 (2017): 9769-9777.

Lafaysse, M., Morin, S., Coléou, C., Vernay, M., Serça, D., Besson, F., Willemet, J.M., Giraud, G., and Durand, Y. (2013). Towards a new chain of models for avalanche hazard forecasting in French mountain ranges, including low altitude mountains. Int. Snow Sci. Workshop Grenoble-Chamonix Mont-Blanc.

MacDonald, M. K., Pomeroy, J. W., and Pietroniro, A. (2009). Parameterizing redistribution and sublimation of blowing snow for hydrological models: tests in a mountainous

subarctic catchment. Hydrological Processes, 23(18), 2570-2583

Marsh, C. B., Pomeroy, J. W., and Spiteri, R. J. (2012). Implications of mountain shading on calculating energy for snowmelt using unstructured triangular meshes. Hydrological Processes, 26(12), 1767-1778.

Pomeroy, J.W., Hedstrom, N. and Parviainen, J. (1999). The snow mass balance of Wolf Creek: effects of snow, sublimation and redistribution. In Wolf Creek Research Basin: Hydrology, Ecology, Environment, Pomeroy JW, Granger R (eds). Environment Canada: Saskatoon; 15–30

Pomeroy, J.W., Bewley, D.S., Essery, R.L.H., Hedstrom, N.R., Link, T., Granger, R.J., Sicart, J.E., Ellis, C.R. and Janowicz, J.R.. (2006). Shrub tundra snowmelt. Hydrological Processes 20: 923–941.

Trujillo, E., Ramírez, J. A., & Elder, K. J. (2007). Topographic, meteorologic, and canopy controls on the scaling characteristics of the spatial distribution of snow depth fields. Water Resources Research, 43(7).

Verfaille, D., Déqué, M., Morin, S. and Lafaysse, M. (2017). The method ADAMONT v1.0 for statistical adjustment of climate projections applicable to energy balance and surface models. Geoscientific Model Development, 10, 4257-4283

Vionnet, V., Six, D., Auger, A., Lafaysse, M., Queno, L., Reveillet, M., Dombrowski-Etchevers, I., Thibert, E. and Dumont, M. (in prep). Influence of precipitation datasets on sub-kilometric simulations of snowpack and glacier winter mass balance in alpine terrain.
* * *

---

## Author Comment (AC3) · 22 Jan 2018

Dear editor, dear reviewers,

We are pleased to submit a revised version of the manuscript. First of all we would like to thank you for your effort in improving the work. Your comments and recommendations have been very helpful and we think that they allowed us to come up with a revised manuscript, which is now better organized and easier to be read.

Looking forward to your reply,

[Figure]

Jesús Revuelto and co-authors.

Please also note the supplement to this comment:
https://www.the-cryosphere-discuss.net/tc-2017-184/tc-2017-184-AC3-supplement.zip

————————————————————